# Contrast Information Dynamics: A Novel Information Measure for Cognitive Modelling

**DOI:** 10.3390/e26080638

**Published:** 2024-07-27

**Authors:** Steven T. Homer, Nicholas Harley, Geraint A. Wiggins

**Affiliations:** 1Computational Creativity Lab, Artificial Research Group, Vrije Universiteit Brussel, 1050 Etterbeek, Belgium; 2Cognitive Science Research Group, School of Electronic Engineering and Computer Science, Queen Mary University of London, London E1 4NS, UK

**Keywords:** information dynamics, cognitive modelling, information theory, information measure, continuous systems, coordinate invariance, specific information, predictive information, Markov process, Gaussian process

## Abstract

We present *contrast information*, a novel application of some specific cases of relative entropy, designed to be useful for the cognitive modelling of the sequential perception of continuous signals. We explain the relevance of entropy in the cognitive modelling of sequential phenomena such as music and language. Then, as a first step to demonstrating the utility of constrast information for this purpose, we empirically show that its discrete case correlates well with existing successful cognitive models in the literature. We explain some interesting properties of constrast information. Finally, we propose future work toward a cognitive architecture that uses it.

## 1. Introduction

*Information dynamics* study how the information provided by a process changes from one moment to the next. They are concerned with the information provided by *specific* observations at *specific* moments in time. While information theory [1] is concerned primarily with the expected behaviour of collections of random variables, information dynamics focus on specific instantiations of stochastic processes. Both the sequentiality and the specificity of a realisation of a stochastic process are central to the study of information dynamics. This focus on specific sequences distinguishes information dynamics from information theory more generally.

A realisation of a stochastic process can be represented as a sequence of observations. In designating a point in the sequence as the current moment in time, the sequence can be partitioned into three temporal regimes: the past (previous observations), present (current observation), and future (subsequent observations). Information dynamics measure how these three regimes inform one another for any designated moment in time. For instance, how much information does the present observation provide about the future, given what was seen in the past? Information dynamics are a natural fit for studying phenomena that evolve over time, but can also be applied in situations where stochastic processes do not have a natural temporal aspect. For example, the process of estimating model parameters from data can be considered a stochastic process when that estimation proceeds in a sequence of steps as more data are observed. Even when the data are independently and identically distributed, when consumed in a sequential manner in the process of estimation, the result is a stochastic process of the estimated parameter.

Previous work on information dynamics [2,3,4,5] focused almost exclusively on discrete-time discrete-state (DTDS) processes with a finite number of states. In order to quantify the information dynamics of a DTDS process, information content and Shannon entropy [6] have been used to predict the cognitive phenomena of the unexpectedness and uncertainty of a given observation in the context of previous observations. However, when working with stochastic processes of a continuous state, neither information content nor Shannon entropy are suited to the task, because both of these measures lack the property of *coordinate invariance* (Section 3.2): their value is dependent on the coordinate system used to describe the process, as shown in Section 3.3. In Section 4, we specify a form of relative entropy called *contrast information* that is not only coordinate invariant, but, like other common information measures, is also non-negative. Contrast information is closely related to information content and Shannon entropy in the discrete case (Section 4.3), but, unlike information content and Shannon entropy, can also be applied to stochastic processes over continuous state spaces. The ability to measure the information dynamics of continuous state processes greatly expands the realm of inquiry that information dynamics can study. Instead of being constrained to only discrete state spaces when using information content and Shannon entropy, contrast information allows for the cognitive modelling of both discrete and continuous state spaces with a single measure of information. The unified approach allows for a direct comparison of the information dynamics of discrete and continuous models, as opposed to using a patchwork of information measures for disparate models.

In Section 5, contrast information is defined with respect to the three temporal regimes: the past, present, and future. Permuting these temporal regimes within the expression of contrast information results in six temporal variants that have different semantics. Predictive contrast information (Section 5.2) is the amount of information that the present provides about the future given the past. It can be thought of as the gain in foresight due to the current observation in light of the past. In cognitive terms, high-predictive contrast information corresponds to observations that elicit a dramatic change in our future estimates or behaviour, whereas low-predictive contrast information means that the current observation does not alter the status quo. Connective contrast information (Section 5.3) is the amount of information that the past provides about the future given the present. It can be thought of as measuring the amount of memory required by the process at the current observation to estimate the future. When it is high, estimates of the future require more past context than just the current observation. When it is low, knowledge of the current observation is mostly sufficient in estimating the future, so the process is nearly memoryless at that point. Therefore, it can also be thought of as the degree to which the process is not Markovian at that point. Reflective contrast information (Section 5.4) is the amount of information that the future provides about the present given the past. Intuitively, it corresponds to the gain in hindsight about the present from knowledge of a future observation. When it is high, the future observation provides an improved explanation of the present. When it is low, knowledge of the future does not aid in making sense of the present any more than knowledge of the past already did. These three forward temporal variants of contrast information are mirrored by three backward variants (Section 5.5). The backward variants have similar interpretations to their forward counterparts, except that, instead of working with a fixed past, the backward variants work with a fixed future.

In the current article, we define contrast information and show that it acts as a unified information measure for both discrete and continuous problems in cognitive modelling, and so shows promise as a successor to previous measures applicable solely in discrete domains. Thus, it allows for the generalisation of discrete models to continuous domains. The structure of this article is as follows. We begin in Section 2 with our theoretical framework for cognitive modelling, the Information Dynamics of Thinking, summarising previous research, and in Section 3, explaining why it needs to be extended into continuous state spaces. Next, in Section 4 and Section 5, we lay out the detail of contrast information and its temporal variants. In Section 6, we perform an empirical comparison between information content and contrast information in the discrete case and show that it correlates well with the measures used in earlier work, and thus would produce similar results if used with those models, and in Section 7, we demonstrate how it can be applied in a few examples of Markov and Gaussian processes in both discrete and continuous time. Finally, in Section 8, we summarise the results and propose future directions for research.

## 2. Theoretical Framework

### 2.1. Motivation: Cognitive Modelling

The motivation of the current work arises from a thread of research on the computational cognitive modelling of sequential perception, mostly of music. Cognitive modelling as a research field lies at the intersection of psychology and artificial intelligence. Computational cognitive models use computational methods to simulate hypothetical models of human or animal cognitive behaviour, so that the hypothesis can be tested, by comparing the outputs of the simulation with empirical studies on the relevant organism. These comparisons can be made behaviourally, by observing human or animal responses, or more objectively by measuring physiological responses, as, for example, in electroencephalography. The benefit of this approach is that it renders very precise and operational hypotheses testable, where they are often difficult or even impossible to test otherwise, often for ethical reasons.

The overarching aim of the work reported here was to underpin a new cognitive model, with which we attempt to generalise a well-established discrete symbolic statistical model of learning and cognitive prediction to a continuous simulation, which we believe will have greater explanatory power. The ultimate aim is to simulate the working of the mind/brain (we use this joint term because it is not yet clear to us where the boundary between the mind and the brain lies, if indeed there is one) at a level of abstraction rather higher than the neural substrate (wetware), but also much more granular than what is readily achievable by observation alone. The purpose of the current section is to explain enough of this background to render the motivation of the subsequent research report clear, and to provide further references for the interested reader.

### 2.2. Predictive Cognition

Over the past decade, there has been a flurry of interest in the idea of *predictive cognition*, which proposes that one key function of cognition is to predict events in the immediate environment of the organism in order to facilitate efficient and effective actions in that environment. Friston [7] proposes the idea of *free energy,* which is formally the variational upper bound of a quantity sometimes called *surprisal* [7] or *information content* [6]. This may be thought of as a measure of *prediction error*, when an organism attempts to predict external events from an internal model of the world; thus, it tells the organism when that model needs to be corrected. Schmidhuber [8] proposes a model of creativity based on the compression of perceptual data and prediction from the compressed model. Clark [9] convincingly presented the philosophical case for this kind of model.

### 2.3. The Information Dynamics of Music and Language

Music is an excellent means by which to study the human mind [10] (Section 3), in particular because it may be considered a(n almost) closed system [11], requiring very little reference outside itself, and, at the same time, demonstrating cognitive effects rather clearly. It is also worthy of study in its own right, because it is hugely important to humans in general, and is (therefore) also an important economic driver in the modern world.

In the computational study of music, the idea of predictive cognition arose rather earlier than in the work cited above, since music perception and appreciation depend heavily on expectation and the fulfilment or denial thereof [12]. Here, information theory appears in two capacities. First, as in the work of Friston [7], it serves to measure how well a learned model fits the data, overall, from which it was learned. Second, in a more granular application, it serves to measure the local information content and entropy [6] for each tone in a musical sequence. The groundwork for this approach was laid by Conklin [13] (see also Conklin and Witten [14]) by providing a straightforward (though not simple) learning mechanism that could cope with the extreme multidimensionality of musical signals. Pearce [2,15] demonstrated that Conklin’s theoretical model is in fact a cognitive model, by implementing it in what has come to be known as IDyOM, short for “Information Dynamics of Music”, and showing that the predictions of the model correlate well (*r* = .91) with human responses, over multiple studies and musical genres. At the time of writing, IDyOM remains the best model of melody perception in the music cognition literature.

IDyOM models human cognition using a complex multidimensional representation constructed from a minimalist data representation of discrete symbols for pitch and musical time: from these basic features, other features are derived, such as pitch contour, scale degree, and metrical structure, which are calculated from the basic features and made available to the model. Each musical event is then represented as a tuple of basic and derived musical features, each of which contributes to a variable-order Markov model, as specified by Conklin and Witten [14] and Pearce [2]. In using the Prediction by Partial Match (PPM) algorithms [16,17], a distribution is obtained for each feature at each point in a melody. These distributions are then combined in an empirically validated model [18] to produce one distribution over musical pitch (again represented discretely). This distribution is shown to correlate well with empirical data from earlier and new studies of musical expectation [19,20]. Furthermore, the information-theoretic outputs of the system, in terms of the information content of observed symbols and entropy of the distributions from which they are drawn [6], have been shown to correlate with human perception at what might be called a meta-level with respect to pitch prediction: information content correlates with the unexpectedness of what is perceived [20], and entropy correlates with uncertainty experienced during perception [21].

In a prequel to Schmidhuber’s model [8], Pearce [2] and Pearce and Wiggins [22] showed how predictions from statistical models can simulate the creation of novel music. This work underlines the need for simulations that can learn the representations that they use, a property that symbolic systems usually lack, and on which the current success of deep learning depends. Similar work in this vein is Deep Music Information Dynamics [23], which takes a comparable approach to the information bottleneck [24], where the learned embeddings of a listener are compressed, yet still predictive, representations of a musical surface with a higher rate of information dynamics.

Notwithstanding its success as a model of a rather specific aspect of music cognition, namely melody perception, IDyOM is restricted by its discrete data representation. Continuously varying percepts, such as portamenti and the complex varying timbres of electronic music and of speech, are beyond its intended range. The new information measures proposed here are necessary to allow the perceptual modelling of these continuous phenomena.

### 2.4. Sequence Segmentation and Boundary Entropy

The information-theoretic outputs of IDyOM can also be used to analyse structure in music, using the detection of *boundary entropy* [25,26]. Boundary entropy formalises the notion of a chunk of information [27] embedded in a sequence. Intuitively, more information is shared *within* a chunk than *between* chunks. For instance, given all the words but one in a sentence, it should be easier to guess that missing word than to guess a word in the following sentence. Alternatively, as one hears the successive words in a sentence, one usually becomes more certain of its ending as it proceeds, and so entropy decreases. When the end of the sentence is reached, one does not know what is coming next, and therefore, uncertainty, as well as entropy, increases. Since there is usually more information shared between words within a sentence than shared between words in different sentences, sentences can naturally be considered as chunks. Boundary entropy detects the boundaries between chunks by measuring this characteristic difference in information between them.

If a sequence of data exhibits sequential regularity of the kind naturally captured by grammar, then both information content and entropy will trend downward on average from the start of a structural unit toward its end, because the available context *within* the structural unit is progressively supplying more information. After the end, there will be an increase in entropy because the statistical correlation across the boundaries between structural units is weaker than that within them, and so less contextual information is available. This principle has been shown to allow the identification of structural boundaries in melodies of various kinds [25], by means of a simple process of peak-picking in the information-theoretic signal: a large peak in the current context tends to coincide with the first symbol in a new structural unit. Sequence processing, as a human capacity, is of course more general than music, and IDyOM has also been used to determine the segmentation of the English language, on one level [28], and hierarchically [29]. The idea of boundary entropy finds support in results (in both music and other sequences), suggesting that the chunks so formed are stored as separate units, thus making them available for subsequent predictions [30,31]. Based on this, Wiggins and Sanjekdar [32] applied the same idea to text, successfully identifying morpheme and word boundaries, while also proposing an entropy-based simulation of memory consolidation that guides model correction by identifying high-information content in the stored model. Segmentation in both music and language naturally connects with the management of the temporal structure, or meter; Agres et al. [33] consider this in the context of the Information Dynamics of Thinking model, outlined below.

In related research, Abdallah and Plumbley [5] considered a broader range of information-theoretic simulations, successfully applying them to a range of artificial musical data. Pearce and Wiggins [3] and Agres et al. [4] presented summaries of the work applied to music.

In summary, there is substantial evidence that the human experience of information quantities, as simulated by Shannon measures [1], plays a direct part in the working of the non-conscious and the conscious mind/brain. In other words, it seems that humans are both consciously and non-consciously sensitive not just to the content of perceptual signals, but also to their information-theoretic properties.

### 2.5. The Information Dynamics of Thinking

The discrete symbolic simulations described above constitute good evidence that information dynamics offer a route to understanding some important aspects of human cognition. Wiggins [10] proposes to generalise these ideas beyond musical stimuli (while still maintaining music as a paramount way into human cognition), and, crucially, to a continuous world model. This proposal views the mind/brain as a collection of interacting oscillatory systems, a view which is increasingly gaining traction in cognitive science (e.g., [34,35,36]). In this proposal, perceptual signals are represented as waves, and therefore, sound is a particularly good starting point: recent research shows that perceived sound and its significant correlates are reproduced as electrical waves in the mind/brain, notwithstanding the fact that it has been deconstructed by the ear prior to its arrival in the auditory cortex [37,38,39].

The Information Dynamics of Thinking (IDyOT) cognitive architecture [26] proposes a cyclic process of learning and segmentation to build a model of continuous percepts that is analogous to the discrete models of IDyOM. A process of categorisation based on the structural similarity of segments allows the same process to be applied repeatedly, building layers of progressively more abstract and less granular representations, as in the work of Wiggins and Sanjekdar [32] (This process is similar to the process of the deep learning of representations, the methodological benefit being that the IDyOT model is driven, bottom-up, and follows scientific theory. In contrast, information loss in the training of deep networks generally renders their operation inscrutable and therefore generally not amenable to scientific enquiry as cognitive models. This issue has spawned the new research subfield of *explainable artificial intelligence*).

Homer et al. [40] show how the lowest layer of an IDyOT model of Western Tonal Music might be based in a Hilbert Space of Resonances, directly representing cognitive representations of musical sound. A Hilbert space of resonances allows for geometrical reasoning about concepts (resonances) in some conceptual space (Hilbert space). For instance, by representing notes or chords as sets of resonances, we can reasonably talk about the distance or angle between notes or chords, relative to the conceptual space in which they are positioned. However, this work does not encompass the need for segmentation, because it focuses on static representations of static percepts. The current article presents a novel method to underpin segmentation in a time-variant version of this and other representational spaces of the same mathematical kind.

In the following sections, we introduce a set of information-theoretic measures that admit the analysis of information dynamics in continuous representations, in a way analogous to information content and entropy in the discrete case outlined above. Given such measures, the way is open to determining boundary entropy and an implementation of segmentation in IDyOT. To ensure that the measures remain consistent across the divide between discrete and continuous representation, we demonstrate a good correlation between the former measures and our new measures applied in the discrete case.

### 2.6. A Simple Taxonomy of Information Measures

In the next section, we will survey other information measures in the literature that have been previously used to study information dynamics. To help us make sense of these related information measures, we will use a simple taxonomy to distinguish between them. There are three sorts of information measures relevant to the information dynamics of a sequence:**Global information measures** characterise the expected behaviour of the process as a whole, independent of time. For instance, the entropy rate [41] (Ch.4) of a stochastic process is the mean rate of growth in entropy as the length of the sequence tends to infinity ($\lim_{n\to \infty }H\left(X^{n}\right)/n$). It provides a global measure of information production associated with the entire stochastic process.**Process information measures** summarise the expected behaviour of a stochastic process relative to specific points in time. For instance, in nonstationary processes, the value of the entropy $H\left(X_{n}\right)$ will depend on the provided time index *n*. It is a process measure because it tracks the expected information produced by a stochastic process over time.**Dynamic information measures** measure the behaviour of specific realisations of a process at specific points in time. For instance, the value of the information content, $-\log p(x_{n})$, will depend not only on the time index *n*, but also the value $X_{n}=x_{n}$ that the process takes on at that time. It is a dynamic measure tracking the information provided by a specific sequence of a stochastic process over time.

These different types of measures are distinguishable by what appears in their expressions. As a rule of thumb, if the measure is independent of time, it is a global information measure; if it is dependent on time but not a specific realisation, it is a process information measure; if it is dependent on both time and a specific realisation, it is a dynamic information measure.

### 2.7. Other Information Measures Related to Information Dynamics

Many other information measures have been proposed to study the information dynamics of a stochastic process. As discussed in Section 2.3 and Section 2.4, contextual information content and contextual entropy have been used to successfully model music and language cognition, as well as to aid in sequence segmentation by means of boundary entropy, using IDyOM [2]. Here, information content is used as a dynamic measure of contextual unexpectedness, and entropy is used as a dynamic measure of contextual uncertainty over a specific sequence. With the information content and entropy measured for each point in the sequence, other global measures can be evaluated, such as the temporal-mean information content or temporal-mean entropy over the entire sequence. Though in IDyOM, entropy is conditioned on a specific previous context, thereby making it a dynamic measure, if we instead measure the expected value conditioned on a random context, entropy would then be a process measure.

Other variants of entropy relevant to stochastic processes may incorporate time into their formalism. Dynamic multivariate entropy [42], is a process measure that is equivalent to the Shannon entropy of a model which changes relative to an explicit time parameter. Adaptive entropy, such as [43], is a broad term referring to how Shannon entropy can affect or be affected by some varying underyling model, which may also be nonstationary or nonergodic. These process measures are particular forms of Shannon entropy tailored to their use case.

An important variant of entropy relevant to information dynamics is Kolmogorov–Sinai (KS) entropy [44], also called metric entropy. KS entropy was proposed as an informational invariant of a dynamical system, akin to the Shannon entropy of a probability distribution or entropy rate of a stochastic process. Instead of thinking of a stochastic process as merely a sequence of random variables, we can consider the sequence as a realisation of a dynamical system [45] that governs the possible paths the process can take. For instance, any autoregressive model [46], such as transformer models [47] or state-space models [48], can be considered a dynamical system, since each successive output is an iterated function of the previous context or states. KS entropy measures the mean rate of creation of information in the system by examining the probability that different regions of its phase space will be inhabited over time. More formally, it is the maximum entropy rate over all finite partitions of the phase space of the dynamical system. Similar to the entropy rate of a stochastic process, KS entropy is a global measure of information associated with the system.

When the dynamical system is not provided a priori, an infinite amount of data would be needed to accurately calculate KS entropy from data. Since we only have access to a finite amount of data, we must approximate KS entropy by finite means [49]. Approximate entropy [50] was proposed as a model-free estimate of the KS entropy of a dynamical system given a finite sequence of data generated from that system. The estimate provided by approximate entropy turns out to be biased, so sample entropy [51] was proposed as an unbiased estimate of KS entropy. When the dynamics of the system operate at multiple time scales, the sample entropy can be calculated for progressively more coarse-grained versions of the process, resulting in multiscale entropy [52]. Like KS entropy, approximate entropy, sample entropy, and multiscale entropy are global measures of information production in the system, but they are also model-free estimators of KS entropy. That is, they make their estimates directly from data without assuming an underlying data-generating model. This differs from KS entropy, which measures the dynamics of a known underlying system.

Other information measures have been formulated primarily in terms of the mutual information between different parts of a stochastic process, such as the past, present, and future. The information rate of Dubnov [53] is defined as the mutual information between the past and present, making it a process measure that characterises the rate of growth in multi-information (a generalised form of mutual information) at each point in the stochastic process. As the time index of the present tends to infinity, the asymptotic information rate becomes a global information measure akin to the entropy rate. Closely related to the information rate is the predictive information of Bialek and Tishby [54], defined as the mutual information between the past and future, but extending only a certain duration into the past and future: they are primarily interested in how the predictive information varies as a function of that duration. Since predictive information could vary depending on where it is centred on the stochastic process, it can be considered a process measure. However, since Bialek and Tishby [54] are primarily interested in stationary processes, predictive information becomes a global information measure because it will take on the same values at each point in the process.

Abdallah and Plumbley [5] proposed multiple related information measures defined as the mutual information between the present and future, given the past. They measure the degree to which the present informs us about the future, given what was already observed in the past. Instantaneous predictive information uses a specific realisation of the past and present, making it a dynamic measure, whereas the predictive information rate measures the expected behaviour over both the past and present, making it a process measure. This and other related work [4,5,55,56] provided the inspiration for our work. In fact, instantaneous predictive information is the same as what we propose in Section 5.2; however, we have generalised and extended the idea into what is presented in the subsequent sections.

Since we are focusing on the information dynamics of specific realisations of stochastic processes at specific points in time, we are interested first and foremost in dynamic information measures. As shown in this section, previous work tends to focus more on process and global measures. In some cases, it is simple to convert a process or global measure into a dynamic measure, as with predictive information rate and instantaneous predictive information, but some familiar identities and inequalities fail to hold when dealing with specific outcomes instead of the expected behaviour, so such a conversion may not be straightforward. Although many of the process and global measures highlighted here enjoy the same desirable properties that we will present in the following sections, they are categorically different from dynamic measures, because they do not deal with specific realisations at specific points in time. As such, we focus our analysis on addressing issues presented by other dynamic information measures, not process and global measures. In particular, information content, Shannon entropy, and differential entropy are some of the primary tools used by dynamic information measures, so we will focus our analysis on these measures. In Section 8.1, we will relate back to some of the process and global measures highlighted here.

## 3. Information Dynamics in Continuous Systems

### 3.1. Motivation

There are a variety of dynamic information measures that can be applied when working with a discrete state space, such as information content or Shannon entropy, and other measures that can be applied when working with a continuous state space, such as differential entropy. Even more possibilities exist when dealing with discrete-time processes or continuous-time processes. Having all of these options can be useful, but the result is a plethora of disparate measures used in different cases. As such, performing a principled comparison between these measures becomes difficult or even arbitrary. Instead of using a different measure of information depending on whether the problem is discrete or continuous, we want a single measure of information that can be used uniformly for all cases, thereby allowing a fair comparison of information dynamics regardless of the discrete or continuous nature of the process in question.

The primary motivation of this paper is to provide a unified information measure for both discrete and continuous state spaces with both discrete and continuous time processes. Dynamic information measures have previously been mostly confined to measuring discrete-time processes over discrete state spaces. The overarching contribution of the current paper is to propose and validate a set of novel information measures that extend the theory of Information Dynamics beyond discrete representations to also encompass continuous ones in a unified manner. To this end, there are a few common-sense criteria that our information measure must fulfil:**Dynamic Measure** The measure must quantify information associated with a specific observation. Since we are concerned with specific realisations of a stochastic process at specific points in time, the information measure must be a dynamic measure.**Uniform Application** The measure must be able to work for both discrete-time and continuous-time stochastic processes over both discrete and continuous state spaces in a uniform manner. When the measure is applied uniformly, we can compare different types of processes in a meaningful way.**Coordinate Invariance** The measure must be independent of the coordinate system used to describe the process. Put another way, different but equivalent descriptions of a process must provide the same amount of information.

### 3.2. Coordinate Invariance

The principle of coordinate invariance [57], also called parameterisation invariance or general covariance, states that Nature does not possess a coordinate system; all coordinate systems are imposed by us to aid in our analysis of Nature. As such, the fundamental quantities of Nature must be invariant to the coordinate system used to describe those quantities, since they exist outside our means to describe them.

For instance, suppose we are measuring the outside temperature each morning. Generally, temperatures do not vary wildly from day to day, so observing today’s temperature gives you some information about what tomorrow’s temperature may be. The temperature itself does not change depending on whether you measure in Fahrenheit or Celsius, so the amount of information provided by those observations must be the same regardless of which way you chose to describe the temperature. Those are just two different coordinate systems describing the same sequence of observations. A measure of information should be coordinate invariant because the information provided by an observation is independent of the coordinate system used in its description. Put another way, though different coordinate systems may be more or less useful or convenient for describing an observation, the choice of coordinate system itself does not carry information.

Note that describing a quantity in a different coordinate system is not the same as describing a quantity at a different level of granularity. To continue with the example, we could describe the temperature as hot, pleasant, brisk, cold, or freezing, instead of measuring it in Fahrenheit or Celsius. The information provided by these categories of temperatures will be different from the information provided by the reading in Fahrenheit or Celsius. They are expressed at a different level of granularity. Since each category corresponds to a range of temperatures, we cannot invert the relationship without ambiguity, whereas the relationship between Celsius and Fahrenheit is invertible. Therefore, describing the temperature with these categories is different from using another coordinate system, and so, in this case, would not be coordinate invariant.

Although there is a larger class of coordinate transformations that can be applied in the discrete case, since we are concerned with the information dynamics of continuous state spaces, we will focus here on differentiable coordinate transformations.

**Definition** **1**(Coordinate Transformation)**.** *A coordinate transformation is a differentiable, invertible function φ:X→Y. That is, for all x∈X and y∈Y, y=φ(x) and $x=\varphi^{-1}\left(y\right)=\psi \left(y\right)$, and $\varphi ,\psi \in C^{1}$, i.e., they both belong to the set of differentible functions $C^{1}$.*

**Definition** **2**(Coordinate Invariance)**.** *A function f is coordinate invariant if and only if for all x∈X and y∈Y, and for all coordinate transformations φ and $\psi =\varphi^{-1}$,*
(1)f(x)=f(φ(x))=f(ψ(y))=f(y).

In short, coordinate invariance just means that we obtain the same result regardless of the coordinate system being used. For us, this is particularly relevant when dealing with probability distributions. For discrete random variables, a coordinate transformation y=φ(x) amounts to relabeling the elements in X with the corresponding elements in Y. So, for probability mass functions $p_{X}\left(x\right)$ and $p_{Y}\left(y\right)$, we have
(2)$$p_{X}\left(x\right)=p_{Y}\left(\varphi \left(x\right)\right)\;\mathrm{for}\;\mathrm{discrete}\;X,Y.$$

As such, any measure of information involving discrete random variables is automatically invariant to coordinate transformations. In particular, information content and Shannon entropy are coordinate invariant when working in a discrete state space.

However, when working with continuous random variables, the picture looks slightly different. In addition to relabeling the elements of X with the corresponding elements of Y, we must also scale the probability density according to the absolute value of the derivative of the coordinate transformation evaluated at *x*. So, for probability density functions $p_{X}\left(x\right)$ and $p_{Y}\left(y\right)$, we have
(3)$$p_{X}\left(x\right)=p_{Y}\left(\varphi \left(x\right)\right)\left|\varphi^{\prime }\left(x\right)\right|\;\mathrm{where}\;\left|\varphi^{\prime }\left(x\right)\right|=\left|\frac{d\varphi }{dx}\left(x\right)\right|\;\mathrm{for}\;\mathrm{continuous}\;X,Y.$$
In higher dimensions with a vector of continuous random variables, we scale according to the determinant of the Jacobian *J* instead.
(4)$$p_{X}\left(x\right)=p_{Y}\left(\varphi \left(x\right)\right)\left|J_{\varphi }x\right|\;\mathrm{where}\;\left|J_{\varphi }x\right|=\det\left(\frac{\partial \varphi_{i}}{\partial x_{j}}\left(x\right)\right)$$

Recall that we are concerned with the information provided by a specific observation. In the discrete case, an observation corresponds to an event to which we can assign a nonzero probability; however, in the continuous case, the probability of a specific observation is always zero, because that observation corresponds to the event that a continuous random variable equals a specific value. Nonetheless, we still want to associate some notion of information with that specific continuous-valued observation. This is often solved by first quantising the observation by partitioning the continuous state space into a discrete state space, and then using a discrete-state measure of information of that quantised observation. However, this introduces another problem on top of the original problem, because the choice of partition will have an effect on the amount of information provided by an observation. Ultimately, quantisation [41] (Ch. 10) is a different issue than the one addressed in this paper, so we will not resort to quantised versions of observations when measuring information in the continuous case.

### 3.3. A Specific Information Measure of Information Dynamics

To reiterate the most relevant parts of Section 3.1, our measure of information dynamics must satisfy three criteria: (1) it measures the information associated with specific observations (at specific points in time); (2) it works with both discrete and continuous state spaces (in both discrete and continuous time); (3) it is also coordinate invariant. Since all measures of information are coordinate invariant in the discrete case, we only need to check if an information measure is coordinate invariant in the continuous case.

Previous work relied on information content and Shannon entropy [6] to measure the information dynamics of a sequence; however, as shown in Theorem A1 (Appendix A), information content $\ell \left(x\right)=-\log p\left(x\right)$ is not coordinate invariant in the continuous case. The expected information content is equal to the Shannon entropy $H\left(Y\right)=-\sum_{y\in Y}p\left(y\right)\log p\left(y\right)$, which can be adapted to measure specific observations by conditioning on that observation, as in $H\left(Y|x\right)=-\sum_{y\in Y}p\left(y|x\right)\log p\left(y|x\right)$. But Shannon entropy is only defined over discrete state spaces, so it is unsuited for the continuous case unless the space is first quantised into a discrete state space, and we have already stated that we are not interested in doing. The continuous analogue of Shannon entropy is differential entropy [41] (Ch. 8), which can also be adapted to a specific observation, as in $H\left(Y|x\right)=-\int_{Y}p\left(y|x\right)\log p\left(y|x\right)dy$. However, like information content, differential entropy is not coordinate invariant, as shown in Theorem A2 (Appendix A). Since information content, Shannon entropy, and differential entropy are not coordinate invariant in the continuous case, we must look to other measures of information to satisfy our criteria.

Mutual information $I\left(Y;X\right)$ measures the amount of shared information between two random variables, or equivalently, the reduction in uncertainty about one variable due to knowledge of another. Though closely related to Shannon entropy and differential entropy, since $I\left(Y;X\right)=H\left(Y\right)-H\left(Y|X\right)$, mutual information is in fact coordinate invariant, a property it inherits from the more general relative entropy, also called KL divergence or discrimination information [58]. But mutual information is not a dynamic measure, since it does not depend on a specific outcome, and so fails our first criterion. To obtain the amount of information provided by a specific observation about another random variable, we instead use specific information $I\left(Y;x\right)$ [59]. The presence of the lowercase *x* in the notation of specific information indicates that a random variable *X* takes on a specific outcome *x*, distinguishing it from mutual information $I\left(Y;X\right)$, which has an uppercase *X* denoting the random variable. This notation is chosen to highlight the link between specific information and mutual information, since mutual information is equal to the expected specific information, $I\left(Y;X\right)=E_{X}\left[I\left(Y;x=X\right)\right]$, in the same way that Shannon entropy is equal to the expected information content, $H\left(X\right)=E_{X}\left[\ell \left(x=X\right)\right]$.

There are two reasonable definitions of specific information that equal mutual information in expectation [60]. One option is defined as a relative entropy $I\left(Y;x\right)=D\left[p(y|x):p(y)\right]$. The other option is defined as an entropy difference G(Y;x)=H(Y)−H(Y|x). When defined as an entropy difference, specific information is additive (Theorem A6 (Appendix A)). The property of additivity implies the chain rule of information: the information gained from observing two events is equal to the information provided by observing one of the events plus the information provided by observing the other event, given that you observed the first event. For example, information content is additive since $\ell \left(x,y\right)=\ell \left(y|x\right)+\ell \left(x\right)$. In fact, DeWeese and Meister [60] proved that the entropy difference formulation is the only definition of specific information that is both additive and whose expected value is equal to mutual information. Unfortunately, the entropy difference formulation is not coordinate invariant (Theorem A5 (Appendix A)). By contrast, when defined as a relative entropy, specific information is coordinate invariant (Theorem A3 (Appendix A)), but then is not additive (Theorem A4 (Appendix A)). Therefore, any definition of specific information whose expected value is equal to mutual information can either be additive or coordinate invariant, but not both. Since we require coordinate invariance, we must choose the relative entropy definition of specific information, thereby sacrificing additivity in the process.

Many information measures are defined in terms of the mutual information between different segments of a stochastic process, such as the information rate of Dubnov et al. [61], the predictive information of Bialek and Tishby [54], and the predictive information rate of Abdallah and Plumbley [56]. Since by definition, the expected specific information is equal to mutual information, these information measures can also trivially be expressed in terms of specific information.

## 4. Contrast Information

### 4.1. Defining Contrast Information

In the previous section, we examined whether some common measures of information satisfy our three criteria for a measure of information dynamics, finding that the relative entropy formulation of specific information fits the bill. We will now define *contrast information*, our new proposed measure for information dynamics, in terms of specific information. In Section 5, we will then refine this general definition into a set of six temporal variants of contrast information that each incorporate the temporal regimes of the past, present, and future in a different way.

**Definition** **3**(Contrast Information)**.** *Given a stochastic process S and index set T, with convex disjoint subsets $T_{A},T_{B},T_{C}\subseteq T$ providing the target $A=\left\{S\left(t\right):t\in T_{A}\right\}$, source $B=\left\{S\left(t\right):t\in T_{B}\right\}$, and context $C=\left\{S\left(t\right):t\in T_{C}\right\}$, the contrast information $I\left(A;b|c\right)$ is the conditional specific information about a target A from observing a specific source B=b in a specific context C=c.*
(5)$$I\left(A;b|c\right)=E\left[\log\frac{p(A|B,C)}{p(A|C)}\;|\;B=b,C=c\right]$$

Though contrast information is a particular form of specific information, and of an order for which *A* and *b* should not matter, we will use the following positional convention to help keep track of things. The target *A* is in the first position to the left of the semicolon which is the random variable over which the expectation is taken. The source *B* is in the second position to the right of the semicolon, which is a specific outcome B=b given in the conditional expectation. The context *C* is in the third position to the right of the vertical bar, a specific context C=c also given in the conditional expectation.

Depending on whether A is a discrete or continuous state space, the expectation in contrast information takes the form of a sum or integral, respectively.
(6)$$\begin{array}{ccc} I\left(A;b|c\right) & =\sum_{a\in A}p\left(a|b,c\right)\log\frac{p(a|b,c)}{p(a|c)}\; & \left(A\;\mathrm{discrete}\right) \end{array}$$
(7)$$\begin{array}{ccc} I\left(A;b|c\right) & =\int_{A}p\left(a|b,c\right)\log\frac{p(a|b,c)}{p(a|c)}da\; & \left(A\;\mathrm{continuous}\right) \end{array}$$

To provide some intuition about contrast information by way of analogy, suppose you are sitting at a red light at a car intersection. After a little while the light turns green; you hit the gas to pass through the intersection. Prior to the light turning green, each moment observing the red light was much like the previous moment. One moment at the red light followed by another is a rather monotonous situation. But when the light turns green, things change dramatically as you hit the gas to move forward. Sitting at the red light and observing the red source in a red context corresponds to a small or even zero value of contrast information, since distinguishing between different moments at the red light is difficult. Once the red light turns green, contrast information increases, because observing the green source in a red context causes a dramatic shift in your behaviour, changing from a stopped car to driving the car forward.

We can formalise the intuition behind this analogy if we (unrealistically) model the stop light as a two-state discrete-time Markov chain, where at any moment in time the probability of remaining the same color is much higher than the probability of changing colors.

**Example** **1.***Suppose we have a two-state Markov chain S with a* 
*RED*  
*state and* 
* 
GREEN
*  
*state. The probability of remaining the same color is .95 and the probability to change colors is .05.*
$$P\left(S_{n+1}=\mathrm{RED}|S_{n}=\mathrm{RED}\right)=P\left(S_{n+1}=\mathrm{GREEN}|S_{n}=\mathrm{GREEN}\right)=\;.95$$
$$P\left(S_{n+1}=\mathrm{RED}|S_{n}=\mathrm{GREEN}\right)=P\left(S_{n+1}=\mathrm{GREEN}|S_{n}=\mathrm{RED}\right)=\;.05$$
*When the light remains red, the contrast information is nearly zero.*

$$I\left(A;b=\mathrm{RED}|c=\mathrm{RED}\right)=0.02$$


*When the light turns from red to green, the contrast information jumps to almost three bits.*

$$I\left(A;b=\mathrm{GREEN}|c=\mathrm{RED}\right)=2.95$$



It is also informative to interpret contrast information in terms of discrimination information, since contrast information is a particular form of specific information, which is itself a particular form of relative entropy, i.e., $I\left(A;b|c\right)=D\left[p(A|b,c):p(A|c)\right]$. Suppose you are sampling from a distribution, and you are trying to decide whether those samples come from the distribution p(A|b,c) or p(A|c). The relative entropy, and therefore contrast information, from p(A|b,c) to p(A|c) can be thought of as the expected amount of information provided by each sample that aids in discriminating [62] in favor of the distribution p(A|b,c) from p(A|c) when p(A|b,c) is the case. If p(A|b,c) is very similar to p(A|c), it is difficult to determine that p(A|b,c) is the case instead of p(A|c). You would need to draw many samples to be confident that p(A|b,c) is the case instead of p(A|c). Since they look so similar, there is little with which to discriminate between them and therefore a small relative entropy from p(A|b,c) to p(A|c). If the distributions look very different, then fewer samples will be necessary in order to discriminate that p(A|b,c) is the case instead of p(A|c).

Since these two distributions differ only in the given source B=b, contrast information measures the degree to which that source affects the target given the context. If it is difficult to discriminate p(A|b,c) from p(A|c), it means that given knowledge of the context, further including knowledge of the source does not affect the distribution of the target much. Contrast information measures this effect by looking solely at the change in the distribution of the target. Even when we have definitely observed a specific source B=b, if the contrast information is small or zero, it would be difficult to say we observed anything happening at all, *looking only at the distribution of the target*. When contrast information increases, observing a specific source is distinguished from the context, since its presence causes p(A|b,c) to look very different from p(A|c).

All three regimes—target, source, and context—are essential components of contrast information. Given the same context, different sources may provide varying amounts of contrast information about the target. Similarly, different contexts given the same source may provide different contrast information about the target. In Section 5, we will examine temporal variants of contrast information, where the temporal ordering of the target, source, and context regimes determines the semantics of each temporal variant of contrast information. Before that, we must highlight the expected variants of contrast information.

### 4.2. Expected Contrast Information

With respect to the target, contrast information is a function of both a specific source and specific context. As such, when the source or context are unknown and represented as random variables, then contrast information is also a random variable. To represent contrast information as a function of a random variable, we write $I\left(A;b=B|c\right)$ for a random source and $I\left(A;b|c=C\right)$ for a random context. The expected value of the contrast information can then be evaluated over either the source or the context. This corresponds to the expected behaviour for a source or context, as opposed to the specific behaviour provided by standard contrast information. When we take the expectation over both the source and context, we simply have the conditional mutual information $I\left(A;B|C\right)$.

**Definition** **4**(Expected Context Contrast Information)**.** *Expected context contrast information is the expected value of the contrast information over the context, leaving the source fixed.*
(8)$$I\left(A;b|C\right)=E\left[\log\frac{p(A|B,C)}{p(A|C)}\;|\;B=b\right]$$

**Definition** **5**(Expected Source Contrast Information)**.** *Expected source contrast information is the expected value of the contrast information over the source, leaving the context fixed.*
(9)$$I\left(A;B|c\right)=E\left[\log\frac{p(A|B,C)}{p(A|C)}\;|\;C=c\right]$$

Note the difference in notation between the contrast information of a random variable, such as $I\left(A;b=B|c\right)$ and $I\left(A;b|c=C\right)$, which is itself a random variable, and the shorthand for expected variants of contrast information over those random variables, such as $I\left(A;B|c\right)$ and $I\left(A;b|C\right)$. When working with contrast information, if the letter is capitalised, the expectation is taken over that variable. If it is lowercase, then it appears as a given in the conditional expectation.

### 4.3. Relationship with Information Content and Shannon Entropy

In the previous sections, we saw how contrast information is closely related to specific information and mutual information. In this section, we will see that in the discrete case, contrast information, information content, and Shannon entropy are also closely linked.

First, the property of non-negativity is widely used in information-theoretic inequalities in order to prove bounds on certain quantities. Like information content and other measures of information, contrast information is also non-negative (Theorem A7 (Appendix B)). In the discrete case, contrast information is not only non-negative, but it has an upper bound of information content (Theorem A8 (Appendix B)), meaning that contrast information is always wedged between zero and the value of information content. Though it is not formulated as an estimator, if we consider of contrast information as estimating information content, then the mean absolute error (MAE) between information content $\ell \left(b|c\right)$ and contrast information $I\left(A;b|c\right)$ is equal to the conditional entropy $H\left(B|A,C\right)$ of the source given the target and context (Theorem A9 (Appendix B)). This conditional entropy is the same as the difference between the entropy $H\left(B|C\right)$ of the source given the context and mutual information $I\left(A;B|C\right)$ between the target and source given the context. This implies that the MAE represents the complexity in the source that is not explainable using the target and context. When the source is a deterministic function of the target and context, the MAE goes to zero. When the source is independent of the target and context, then the MAE attains its maximum value of $H\left(B\right)$ (Theorem A10 (Appendix B)).

Though contrast information is not defined to be an estimator, we see that it is related to information content through the entropy of the source; however, in the context of information dynamics, the important relationship between information content and contrast information is not in terms of error between the two, but in terms of their correlation, since we are generally concerned with the shape, not the magnitude, of the information signal when studying information dynamics. We perform an empirical investigation of the correlation between information content and contrast information as well as the Shannon entropy and expected contrast information in Section 6.1.

## 5. Temporal Variants of Contrast Information

### 5.1. Temporal Regimes

In order to analyse how information changes in time, it is useful to partition the stochastic process *S* into three temporal regimes represented as random variables: past *X*, present *Y*, and future *Z*. A discrete version of this partition was proposed by [5], which is generalised here to include stochastic processes in continuous time.

These three regimes can be considered a window centred at the present. Information dynamics are then measured by considering successive moments in time as the present, with the past and present relative to each moment. The present is always considered a single moment in time; however, the past and future can be interpreted in several ways, relative to the present. The most general is to consider the abstract or infinite past and future, where the past *X* stands for the entirety of the stochastic process occurring before the present *Y*, and the future *Z* stands for the entirety of the process occurring after the present.
$$\overset{X}{\overset{︷}{\ldots ,S_{n-3},S_{n-2},S_{n-1}}},\overset{Y}{\overset{︷}{S_{n}}},\overset{Z}{\overset{︷}{S_{n+1},S_{n+2},S_{n+3},\ldots }}$$

We can also consider the near past $X^{j}$ and near future $Z^{k}$, which are bounded intervals occurring before and after the present *Y* with the length of those intervals indicated by the superscripts *j* and *k*.
$$\ldots ,\overset{X^{j}}{\overset{︷}{S_{n-j},\ldots ,S_{n-2},S_{n-1}}},\overset{Y}{\overset{︷}{S_{n}}},\overset{Z^{k}}{\overset{︷}{S_{n+1},S_{n+2},\ldots ,S_{n+k}}},\ldots$$

Further, we can consider a past point $X_{j}$ and future point $Z_{k}$, which are specific moments in time occurring at time *j* before and time *k* after the present, respectively.
$$\ldots ,\overset{X_{j}}{\overset{︷}{S_{n-j}}},\ldots ,S_{n-2},S_{n-1},\overset{Y}{\overset{︷}{S_{n}}},S_{n+1},S_{n+2},\ldots ,\overset{Z_{k}}{\overset{︷}{S_{n+k}}},\ldots$$

Different variants of contrast information can be derived by assigning the past, present, and future to the context, source, and target of contrast information. In this way, we can permute the temporal regimes corresponding to the source, target, and context to arrive at different variants of contrast information that provide insight into the structure and dynamics of the sequence in a variety of ways. We explore these temporal variants in the following sections.

The benefit of having flexible notions of past and future is that it may not always be possible to evaluate the entire past or future, whereas evaluating the bounded past and future may be more straightforward. In other cases, measuring the information dynamics using the entire past and future may provide less insight than using the point past and future. The flexibility of defining the past and future allows contrast information analysis to conform to the problem at hand or the data available. The different ways in which to express the past and future are summarised in Table 1.

A discrete-time process can often be considered a uniformly sampled version of a continuous-time process. However, more generally, observations of a continuous-time process are not required to be uniformly sampled in time. The observations may be spread heterogeneously over an interval, or may be observed continuously, and therefore cannot be treated as a discrete-time process in the same way. Previous work in information dynamics focused on DTDS processes, always measuring the information provided by the current observation in the context of the immediate past. Something similar is still possible with continuous-time processes, but we must be careful when defining what is meant by the past context, since those observations may be heterogeneously distributed in time or continuously observed.

We now examine the three forward variants of contrast information: predictive, connective, and reflective contrast information.

### 5.2. Predictive Contrast Information

Predictive contrast information, which is equivalent to the instantaneous predictive information of Abdallah and Plumbley [5], measures the degree to which the future changes after incorporating knowledge of the present.

**Definition** **6**(Predictive Contrast Information)**.** *Predictive contrast information measures the information gained about the future target (A=Z) from observing the specific present source (b=y) in the specific past context (c=x).*
(10)$$I\left(Z;y|x\right)=E\left[\log\frac{p(Z|X,Y)}{p(Z|X)}\;|\;X=x,Y=y\right]$$

Since it is equivalent to the relative entropy from p(Z|x,y) to p(Z|x), it measures the degree to which you can distinguish p(Z|x,y) from p(Z|x) when p(Z|x,y) is the case. So, we have just observed the specific present *y*, which has some effect on the future *Z*. The present state of the world is represented as p(Z|x,y) and the previous state of the world is represented as p(Z|x). Predictive contrast information is then the number of extra bits needed to describe the current state of the world in terms of the previous state of the world. As such, when this quantity is low, incorporating the present does not change the distribution much over what it looked like using only the past: the future has not changed due to observing the present. When this quantity is high, incorporating the present dramatically changes the distribution of the future over what it was before observing the present: the present provides a lot of information about the future.

### 5.3. Connective Contrast Information

Connective contrast information indicates the degree to which the past informs the future more than the present already does. Similar to a memory gap [63], it measures the degree to which knowledge of the present separates the past and the future. As such, it can also be considered as measuring how non-Markovian the process is at the present moment in time. For instance, for Markov processes, all information in the past about the future is contained in the present, so connective information will always be zero for Markov processes.

**Definition** **7**(Connective Contrast Information)**.** *Connective contrast information measures the information gained about the future target (A=Z) from observing a specific past source (b=x) in the specific present context (c=y).*
(11)$$I\left(Z;x|y\right)=E\left[\log\frac{p(Z|X,Y)}{p(Z|Y)}\;|\;X=x,Y=y\right]$$

Connective contrast information measures the degree to which the present aids or hinders the past in informing the future. If most information about the future is contained in the present, then this quantity will be low. When this quantity is high, the present serves as a conduit of information from the past to the future.

### 5.4. Reflective Contrast Information

Reflective contrast information measures how knowledge of the future impacts the present in the context of the past. It can be thought of as measuring hindsight, since the specific future source occurs after the target present. Intuitively, sometimes the significance of an event is only appreciated after the fact, where the outcomes of subsequent events make clear the importance of some previous event. Though not a perfect analogy, suppose that today corresponds to the future and yesterday corresponds to the present. If knowing what you know today would change what you did yesterday, then there would be high reflective contrast information. If you would not change a thing, then there is low reflective contrast information.

**Definition** **8**(Reflective Contrast Information)**.** *Reflective contrast information measures the information gained about the present target (A=Y) from observing a specific future source (B=z) in the specific past context (C=x).*
(12)$$I\left(Y;z|x\right)=E\left[\log\frac{p(Y|X,Z)}{p(Y|X)}\;|\;X=x,Z=z\right]$$

This quantity is low when knowledge of the future does not change the present in the context of the past. The lower the reflective contrast information, the more that information flows in a single direction from the past to the future. When this is high, knowledge of the future provides a lot of information about the present, more than what the past already provided.

### 5.5. Backward Temporal Variants

The three previous variants are considered as variants of forward contrast information because the temporal regime associated with the context occurs before the temporal regime associated with the target. When the context occurs after the target instead, then we have the three backward variants of contrast information. Each of the backward variants is paired with its forward variant by interchanging the appropriate regimes. Formally, the backward variants simply reverse the arrow of time, and measure the forward contrast information in this reversed universe. More intuitively, the backward variants work by fixing some future outcome as certain to occur, and then measuring the contrast information from there. Cognitively, it may seem weird to fix a specific future outcome, since we have not yet experienced that future; however, we frequently operate in this manner in our daily lives whenever we have a scheduled event in the future. For example, suppose that you have a dentist appointment later in the day. Your day will likely be structured differently than usual, and certain events prior to the appointment may carry more or less information than they would on a normal day without the appointment. The dentist appointment corresponds to a fixed specific future outcome.

**Definition** **9**(Backward Predictive Contrast Information)**.** *Backward predictive contrast information measures the information gained about the past target (A=X) from observing the specific present source (B=y) in the specific future context (C=z).*
(13)$$I\left(X;y|z\right)=E\left[\log\frac{p(X|Y,Z)}{p(X|Z)}\;|\;Y=y,Z=z\right]$$

**Definition** **10**(Backward Connective Contrast Information)**.** *Backward connective contrast information measures the information gained about the past target (A=X) from observing a specific present source (B=y) in the specific future context (C=z).*
(14)$$I\left(X;z|y\right)=E\left[\log\frac{p(X|Y,Z)}{p(X|Y)}\;|\;Y=y,Z=z\right]$$

**Definition** **11**(Backward Reflective Contrast Information)**.** *Backward reflective contrast information measures the information gained about the present target (A=Y) from observing a specific past source (B=x) in the specific future context (C=z).*
(15)$$I\left(Y;x|z\right)=E\left[\log\frac{p(Y|X,Z)}{p(Y|Z)}\;|\;X=x,Z=z\right]$$

### 5.6. Terminology and Novelty

Since contrast information is just one form of specific information, you might wonder why we insist on calling it contrast information at all. Why introduce new terminology when contrast information is exactly specific information?

First, the random variables and specific outcomes found in contrast information always correspond to three temporal regimes, and not just any three random variables. The sequentiality of these variables is at the very core of the semantics of contrast information. If we are instead referring only to specific information, there is no assumption of sequentiality, and so the variables present in the expression do not carry temporal semantics in the same way. By referring to contrast information instead, we are indicating that the quantities involved are parts of a process. Since sequentiality is so important in information dynamics, it makes sense to use terminology and notation that incorporates sequentiality as well. This is in line with other information measures surveyed in Section 2.7. Most are defined using mutual information or Shannon entropy, but still deserve a separate name since their temporal semantics distinguish them from those more general concepts.

The second reason is more heuristic. Since in the context of contrast information we always indicate the past, present, and future with specific variables names, X,Y, and *Z*, we gain a lot of notational utility by distinguishing contrast information from specific information. Whenever you see a see *Z* in regard to contrast information, you immediately know that it refers to the future, whereas more generally with specific information, a random variable named *Z* does not necessarily carry that meaning, and it would have to be specified explicitly with each usage to avoid confusion.

Perhaps a more pressing critique concerns the novelty of contrast information. Other global and process information measures based on mutual information already satisfy the criteria for uniform application and coordinate invariance, yet they do not satisfy the requirement of a dynamic measure. By contrast, other dynamic measures like information content and contextual entropy have already been used successfully in measuring information dynamics, but we have shown they are not coordinate invariant. Contrast information satisfies all three criteria, but still leaves room within the formalism to define a set of expected and temporal variants. Though forward predictive contrast information has been previously studied [4,56] as instantaneous predictive information, the connective, reflective, and backward variants of contrast information are novel measures that together form a comprehensive toolbox that can be applied across the board. Not only does contrast information provide dynamic measures for both discrete-time and continuous-time stochastic processes over discrete and continuous state spaces, it also allows us to view a variety of aspects of that process by utilizing the different variants, thereby providing a richer, multi-faceted view of the information dynamics of a sequence. In the next section, we will use this toolbox to demonstrate how contrast information can potentially work with existing cognitive models.

## 6. Contrast Information in IDyOMS

In this section, we use IDyOMS (https://github.com/nick-harley/Idyoms), our new implementation of the IDyOM cognitive model, to apply both the previous information measures of information content and entropy and the corresponding new ones of contrast information to discrete music data. We examine correlations (Pearson and Spearman) between the discrete-state information measures and our newly proposed ones, applied to the same data. On the basis of this satisfactory correlation, we propose that using our more widely applicable measure in place of the more limited previous measures will not only work with IDyOMS and other similar cognitive models when dealing with discrete data but also allow these models to address problems in the continuous domain. Next, we briefly introduce IDyOMS, and then discuss the validation study.

### 6.1. Information Dynamics of Multidimensional Sequences (IDyOMS)

The Information Dynamics of Multidimensional Sequences (IDyOMS) is an implementation, and generalisation, of the Information Dynamics of Music (IDyOM) [2] for use with any kind of discrete multidimensional sequence data: sequences of points in multidimensional feature space, each represented by a finite number of discrete feature values. IDyOMS determines the information content of each sequence event using Markov models of varying orders (up to a specified order bound) for a chosen combination of features (dimensions). Probability distributions over a given dimension are computed using Prediction by Partial Matching (PPM) [64] for each sequence event. These individual feature models, referred to as viewpoint models [14], are then combined according to their predictive power (the entropy of the distribution) to obtain the overall information dynamics of a sequence in terms of evolving information content and entropy.

### 6.2. Method

As previously discussed, such variable order, multiple viewpoint systems are strong cognitive models of unexpectedness, uncertainty, and chunking in the perception of discrete musical sequences. With the ultimate goal of generalising these models for use with equivalent continuous representations using contrast information, we substituted information content for contrast information in IDyOMS and compared the resulting information dynamic profiles of a corpus of 152 folk melodies from Nova Scotia, Canada, employing escape method C, backoff smoothing, and no update exclusion, for order bounds 1 through 10 in the underlying PPM algorithm (datasets, information profiles, and implementations can be shared upon request). These melodies comprise sequences of events, each represented by their pitch and duration. We examined viewpoint models for these features (CPITCH and DUR) separately, as well as a combined model (referred to as the linked viewpoint model [18,65] CPITCH × DUR).

We chose to analyse the correlation between information content and contrast information instead of incorporating contrast information directly into IDyOMS, since incorporating contrast information into IDyOMS would introduce a plethora of issues, issues like how to PPM would work with continuous-time/state processes, since it was designed for use with discrete-time/state processes; or how other PPM-specific concepts like order bounds, smoothing, and update exclusion work when dealing with both past and future contexts, which are ultimately irrelevant to our purposes in this work. Ultimately, incorporating contrast information into IDyOMS would pose more questions than it would answer, and those questions would not be particularly relevant anyway, so we instead measured correlation is an indicator of compatibility, and avoid those questions altogether.

Pearson (ρ) and Spearman ($r_{s}$) correlations were calculated between information content and forward predictive contrast information, and between entropy and expected forward predictive contrast information for the three viewpoint models (CPITCH, DUR, and CPITCH × DUR) for a range or order bounds (0–10).

The two correlations tell us different things, both useful. Pearson correlation estimates overall similarity in the shapes of information profiles. Spearman correlation indicates the degree to which the information profiles exhibit peaks in the same locations. This is highly desirable if contrast information is to be used in place of information content and entropy in future work on segment boundary detection, because, as mentioned above, these methods work by detecting peaks in dynamic information measures.

### 6.3. Results

Table 2 compares information content with forward predictive contrast information. Table 3 compares entropy with expected forward predictive contrast information, showing both forms of correlation for models of order 0 to 10.

The correlations vary across both viewpoint models and order bounds, tending to increase as the order increases. The tables include correlations of the zeroth-order models (static models that do not use past events to estimate probabilities), where, respectively, the entropy and expected forward predictive contrast information are constant (so Spearman does not apply) while the information content and forward predictive contrast information reflect the maximum likelihood of a single feature value occurring in the corpus. Because of the large dataset, even small correlations are statistically significant. Intuitively, we understand that increasing the order bound (the maximum number of past events considered in the prediction) has the effect of decreasing the relative contribution to the information profiles of the distribution over future events, and so contrast information approaches information content.

Correlations were higher for the simple viewpoint models CPITCH and DUR and lower for the linked viewpoint model CPITCH × DUR. This is due to the increased alphabet size of the linked (Cartesian product) feature value domain relative to the corpus size, and the resultant lower probabilities and higher contribution of the distribution over the future. It should be noted that due to the way that PPM generates a distribution over the present given the past, the same algorithm cannot be trivially adapted to generate joint distributions involving the future. In other words the correlations are being taken between the different measures of *subtly different* models. However, we argue that these models are similar enough to make the argument that contrast information has comparable value as an information dynamic measure in cognitive models.

Overall, these results show that in many cases, using contrast information in place of information content and entropy in IDyOMS would not greatly impact the downstream cognitive modelling methodologies for segment boundary detection. However, this is clearly highly dependent on the viewpoint models and order bounds used. Because the major difference between the two methods for computing information profiles is the contribution of distributions over future events, the subsequent application of IDyOMS for cognitive modelling using contrast information should take into account the order bound and alphabet size of a viewpoint model relative to the size of the corpus.

Note that this demonstration is independent of our use of IDyOMS. Any cognitive model based on information dynamics should be able to use contrast information in place of information content and entropy, so we could equally have used a different cognitive model. In fact, any method concerned with measuring the information over time of a stochastic process can make use of contrast information dynamics, not just cognitive models.

## 7. Constrast Information of Some Stochastic Processes

In the previous section, we saw that, empirically, contrast information can serve in place of information content and entropy in information dynamic-based models of cognition, with the important benefit of also being able to work with continuous data. In this section, we will demonstrate how contrast information can be used with four common types of stochastic process. For stochastic processes over a discrete state space, we will analyse the contrast information of a Markov process in both discrete and continuous time. Then, for a stochastic process over a continuous state space, we will investigate the contrast information of a Gaussian process, again in both discrete and continuous time. We will derive closed-form expressions for the contrast information associated with these processes and demonstrate how the forward temporal variants behave when provided a musical melody as a sequence.

This section is meant to provide a taste of how contrast information can be derived and used for some useful classes of stochastic processes. The four examples in the next four subsections were chosen to be illustrative of the affordances of contrast information, not necessarily as good cognitive models of each perceptual sequence. The discrete-state processes use MIDI notes, which are taken as discrete, categorical data, whereas the continuous-state processes use fundamental frequencies determined by converting those MIDI note numbers to Hertz. Even though they are derived from discrete data, these frequencies are still considered continuous data, because although there are only a finite number of different values owing to their origin as a finite set of MIDI notes, the frequencies are nonetheless continuous-valued quantities. This allows us to use the MIDI note representation with Markov processes and the fundamental frequency representation with Gaussian processes, so that all four examples are tracking the same sort of sequence in four different ways.

These examples are not required to understand the larger picture, and are meant to serve as a springboard for those who wish to investigate contrast information in other settings, so the reader may safely skip this section if desired.

### 7.1. Contrast Information of a Discrete-Time Markov Process

A discrete-time Markov Chain (DTMC) is a discrete-time stochastic process over a finite set of discrete states possessing the Markov property. In our case, the future is independent of the past given the present.
$$P\left(S_{n+1}|S_{n},S_{n-1},\ldots ,S_{1},S_{0}\right)=P\left(S_{n+1}|S_{n}\right)$$

When the DTMC is stationary, the transition probabilities between states remained fixed over time, and the process can be represented as a transition matrix *P*. The entry $P_{jk}$ indexed by row *j* and column *k* represents the conditional probability $P_{jk}=P\left(S_{n+1}=j|S_{n}=k\right)$ that, given you are in state *k*, you will transition to state *j* in the next time step, with each column summing to 1, i.e., $\langle 1|P=\langle 1|$, where $\langle 1|$ is a row vector of ones. (Here, we represent the transitions using a right stochastic matrix, though it is also common to see a left stochastic matrix representing the transition matrix. We use the right stochastic matrix representation so that we can use Dirac notation with the standard semantics, where a column vector corresponds to a state, as opposed to row vector with the left stochastic matrix.) When *S* is ergodic, aperiodic, and irreducible, we can also represent the stationary distribution $P(S_{n})$ as the vector $|\pi \rangle =P|\pi \rangle$ so that the $k^{th}$ entry of $|\pi \rangle$ is the stationary probability $\pi_{k}=P\left(S_{n}=k\right)$. The reverse transition matrix *R* has entries $R_{jk}=P\left(S_{n}=j|S_{n+1}=k\right)$ representing the probability that, given you are in state *k*, you just transitioned from state *j*. The entries of the reverse transition matrix can be determined from the forward transition matrix *P* and stationary distribution $|\pi \rangle$, as $R_{jk}=\frac{\pi_{j}}{\pi_{k}}P_{kj}$, or in matrix form, $R=D_{\pi }P^{⊤}D_{\pi }^{-1}$, where $D_{\pi }=\mathrm{diag}\left(|\pi \rangle \right)$.

By selecting the appropriate temporal regimes for *X*, *Y*, and *Z*, we can use the transition matrices *P* and *R* to calculate all the variants of contrast information. Consider the regimes *j* steps into the past and *k* steps into the future.
(16)$$X_{j}=S_{n-j},\;Y=S_{n},\;Z_{k}=S_{n+k}$$

We can express the (j,k)-step contrast information as follows, where $P^{n}$ is the transition matrix *P* multiplied *n* times, and $|x\rangle$ is a one-hot vector representation of the state *x*.
(17)$$\begin{array}{cc} I\left(Z_{k};y|x_{j}\right) & =\sum_{z\in Z}\langle z|P^{k}|y\rangle \log\frac{\langle z|P^{k}|y\rangle }{\langle z|P^{j+k}|x\rangle }\;\;\; \end{array}$$
(18)$$\begin{array}{cc} I\left(X_{j};z_{k}|y\right) & =0\;\;\;\;\;\;\;\;\;\;\;\;\; \end{array}$$
(19)$$\begin{array}{cc} I\left(Y;z_{k}|x_{j}\right) & =\sum_{y\in Y}\frac{\langle z|P^{k}|y\rangle \langle y|P^{j}|x\rangle }{\langle z|P^{j+k}|x\rangle }\log\frac{\langle z|P^{k}|y\rangle }{\langle z|P^{j+k}|x\rangle } \end{array}$$
(20)$$\begin{array}{cc} I\left(X_{j};y|z_{k}\right) & =\sum_{x\in X}\langle x|R^{j}|y\rangle \log\frac{\langle x|R^{j}|y\rangle }{\langle x|R^{j+k}|z\rangle }\;\;\; \end{array}$$
(21)$$\begin{array}{cc} I\left(Z_{k};x_{j}|y\right) & =0\;\;\;\;\;\;\;\;\;\;\;\; \end{array}$$
(22)$$\begin{array}{cc} I\left(Y;x_{j}|z_{k}\right) & =\sum_{y\in Y}\frac{\langle x|R^{j}|y\rangle \langle y|R^{k}|z\rangle }{\langle x|R^{j+k}|z\rangle }\log\frac{\langle x|R^{j}|y\rangle }{\langle x|R^{j+k}|z\rangle } \end{array}$$

The forward predictive and reflective contrast information are illustrated in Figure 1 for a musical melody.

### 7.2. Contrast Information of a Continuous-Time Markov Process

A continuous-time Markov chain (CTMC) [66] is the continuous-time version of a DTMC, where future is independent of the past given the present. When the CTMC is stationary, the transition matrix *P* after some (scalar) time *t* can be expressed as the matrix exponential of a rate matrix *Q*.
(23)$$P\left(t\right)=e^{tQ}$$

Like the transition matrix *P* for the DTMC, the rate matrix *Q* determines the behaviour of a stationary CTMC, with $Q_{j,k}\geq 0,\forall j\neq k$ and $\sum_{j}Q_{jk}=0$. The stationary distribution $|\pi \rangle$ of the CTMC obeys $Q|\pi \rangle =|0\rangle$ and allows us to determine the reverse rate matrix $Q^{\prime }=D_{\pi }Q^{⊤}D_{\pi }^{-1}$, where $D_{\pi }=\mathrm{diag}\left(|\pi \rangle \right)$.

By selecting temporal regimes *X*, *Y*, and *Z* as points, we can use the rate matrices *Q* and $Q^{\prime }$ to calculate all the variants of contrast information. Consider the regimes with time *u* into the past and time *v* into the future.
(24)X=S(t−u),Y=S(t),Z=S(t+v)

We can express the *u* and *v* contrast information as follows, where $e^{tQ}$ is the matrix exponential, and $|x\rangle$ is a one-hot vector representation of the state *x*.
(25)$$\begin{array}{cc} I\left(Z_{v};y|x_{u}\right) & =\sum_{z\in Z}\langle z|e^{vQ}|y\rangle \log\frac{\langle z|e^{vQ}|y\rangle }{\langle z|e^{(u+v)Q}|x\rangle }\;\;\; \end{array}$$
(26)$$\begin{array}{cc} I\left(X_{u};z_{v}|y\right) & =0\;\;\;\;\;\;\;\;\;\;\;\;\;\; \end{array}$$
(27)$$\begin{array}{cc} I\left(Y;z_{v}|x_{u}\right) & =\sum_{y\in Y}\frac{\langle z|e^{vQ}|y\rangle \langle y|e^{uQ}|x\rangle }{\langle z|e^{(u+v)Q}|x\rangle }\log\frac{\langle z|e^{vQ}|y\rangle }{\langle z|e^{(u+v)Q}|x\rangle } \end{array}$$
(28)$$\begin{array}{cc} I\left(X_{u};y|z_{v}\right) & =\sum_{x\in X}\langle x|e^{uQ^{\prime }}|y\rangle \log\frac{\langle x|e^{uQ^{\prime }}|y\rangle }{\langle x|e^{\left(u+v\right)Q^{\prime }}|z\rangle }\;\;\; \end{array}$$
(29)$$\begin{array}{cc} I\left(Z_{v};x_{u}|y\right) & =0\;\;\;\;\;\;\;\;\;\;\;\;\;\; \end{array}$$
(30)$$\begin{array}{cc} I\left(Y;x_{u}|z_{v}\right) & =\sum_{y\in Y}\frac{\langle x|e^{uQ^{\prime }}|y\rangle \langle y|e^{vQ^{\prime }}|z\rangle }{\langle x|e^{\left(u+v\right)Q^{\prime }}|z\rangle }\log\frac{\langle x|e^{uQ^{\prime }}|y\rangle }{\langle x|e^{\left(u+v\right)Q^{\prime }}|z\rangle } \end{array}$$

The forward predictive and reflective contrast information are illustrated in Figure 2 for a musical melody.

### 7.3. Discrete-Time Gaussian Process

A Gaussian process is a stochastic process [46] where all finite-dimensional marginals are multivariate Gaussian distributions. A multivariate Gaussian *S* is defined according to its mean vector μ and covariance matrix Σ,
(31)S∼N(μ,Σ)
and has probability density p(s) when |Σ|>0, i.e., Σ is symmetric positive definite.
(32)$$p\left(s\right)=\exp\left(-\frac{1}{2}\langle s-\mu |\Sigma^{-1}|s-\mu \rangle \right)/\sqrt{{\left(2\pi \right)}^{n}\left|\Sigma \right|}$$

The general formula for the contrast information of Gaussian processes can be derived from the KL divergence between two (multivariate) Gaussians [67], over the target *A* with length *n*.
(33)$$\begin{array}{cc} I\left(A;b|c\right) & =D\left[p(A|B=b,C=c):p(A|C=c)\right] \end{array}$$
(34)$$\begin{array}{cc} \; & =D\left[N\left(\mu_{A|B,C},\Sigma_{A|B,C}\right):N\left(\mu_{A|C},\Sigma_{A|C}\right)\right]\\ & =\frac{1}{2}(\langle \mu_{A|C}-\mu_{A|B,C}|\Sigma_{A|C}^{-1}|\mu_{A|C\rangle }-\mu_{A|B,C} \end{array}$$
(35)$$\begin{array}{cc} \; & \;+\mathrm{tr}\left[\Sigma_{A|C}^{-1}\Sigma_{A|B,C}\right]-\ln\frac{|\Sigma_{A|B,C}|}{|\Sigma_{A|C}|}-n) \end{array}$$
where *S* is partitioned into target *A*, source *B*, and context *C*, with associated mean vector μ and block covariance matrix Σ.
(36)$$S=\left[\begin{array}{c} A\\ B\\ C \end{array}\right],\;\mu =\left[\begin{array}{c} \mu_{A}\\ \mu_{B}\\ \mu_{C} \end{array}\right],\;\Sigma =\left[\begin{array}{ccc} \Sigma_{AA} & \Sigma_{AB} & \Sigma_{AC}\\ \Sigma_{BA} & \Sigma_{BB} & \Sigma_{BC}\\ \Sigma_{CA} & \Sigma_{CB} & \Sigma_{CC} \end{array}\right]$$

The conditional distribution of the target *A* given the source *B* and context *C* is also a multivariate Gaussian, with the following mean vector and covariance matrix.
(37)$$A|B,C∼N(\mu_{A|B,C},\Sigma_{A|B,C})$$
(38)$$\mu_{A|B,C}=\mu_{A}+\left[\begin{array}{cc} \Sigma_{AB} & \Sigma_{AC} \end{array}\right]{\left[\begin{array}{cc} \Sigma_{BB} & \Sigma_{BC}\\ \Sigma_{CB} & \Sigma_{CC} \end{array}\right]}^{-1}\left[\begin{array}{c} B-\mu_{B}\\ C-\mu_{C} \end{array}\right]$$
(39)$$\Sigma_{A|B,C}=\Sigma_{AA}-\left[\begin{array}{cc} \Sigma_{AB} & \Sigma_{AC} \end{array}\right]{\left[\begin{array}{cc} \Sigma_{BB} & \Sigma_{BC}\\ \Sigma_{CB} & \Sigma_{CC} \end{array}\right]}^{-1}\left[\begin{array}{c} \Sigma_{BA}\\ \Sigma_{CA} \end{array}\right]$$

Similarly, the conditional distribution of the target *A* given only the context *C* is also multivariate Gaussian, with a corresponding mean vector and covariance matrix.
(40)$$A|C∼N(\mu_{A|C},\Sigma_{A|C})$$
(41)$$\mu_{A|C}=\mu_{A}+\Sigma_{AC}\Sigma_{CC}^{-1}\left(C-\mu_{C}\right)$$
(42)$$\Sigma_{A|C}=\Sigma_{AA}-\Sigma_{AC}\Sigma_{CC}^{-1}\Sigma_{CA}$$

For stationary discrete-time Gaussian processes (DTGP), the covariance matrix can be expressed in terms of a discrete-time autocovariance function γ(n), i.e., each entry in the covariance matrix $\Sigma_{jk}=\gamma \left(\left|j-k\right|\right)$.

In defining the temporal regimes as extended *j* units of time into the past and *k* units of time into the future, the temporal regimes can be plugged in to the context, source, and target, to yield the different desired variants of contrast information.
(43)$$X^{j}=\left\{S_{n-j},\ldots ,S_{n-1}\right\},\;Y=S_{n},\;Z^{k}=\left\{S_{n+1},\ldots ,S_{n+k}\right\}$$

The forward predictive, reflective, and connective contrast information are illustrated in Figure 3 for a musical melody.

### 7.4. Continuous-Time Gaussian Process

Continuous-time Gaussian processes (CTGPs) [68] are essentially the same as DTGPs, except that a covariance operator or kernel function may be used to deal with the continuity of the temporal domain. When the CTGP is stationary, then we can use a continuous-time autocovariance function γ(t) to describe its covariance, as opposed to the discrete-time autocovariance used for DTGPs.

Suppose we define the source *B* and target *C* as points in time. This may be useful when we have heterogeneously sampled data that are not continuously observed. In this case, the temporal regimes must all be associated with single points in time,
(44)$$X_{u}=S\left(t-u\right),\;Y=S\left(t\right),\;Z_{v}=S\left(t+v\right)$$
and the contrast information for Gaussian processes simplifies to the following.
(45)$$\begin{array}{cc} I\left(A;b|c\right) & =D\left[p(A|B=b,C=c):p(A|C=c)\right] \end{array}$$
(46)$$\begin{array}{cc} \; & =\frac{1}{2}\left(\frac{{\left(\mu_{A|B,C}-\mu_{A|C}\right)}^{2}}{\sigma_{A|C}^{2}}+\frac{\sigma_{A|B,C}^{2}}{\sigma_{A|C}^{2}}-\ln\frac{\sigma_{A|B,C}^{2}}{\sigma_{A|C}^{2}}-1\right) \end{array}$$

The different variants of contrast information can be determined by plugging in the relevant temporal regimes. The forward predictive, reflective, and connective contrast information are illustrated in Figure 4 for a musical melody.

## 8. Discussion

### 8.1. Comparison with Other Information Measures

In Section 2.7, we saw how information dynamics can be measured using a variety of other global, process, and dynamic information measures. Now that we have presented contrast information and its temporal variants, and we can compare it with previous measures. But first, we must make an important clarification. Contrast information is not a model, estimate, or algorithm; it is solely a measure of the information dynamics of a sequence. Contrast information measures the information dynamics relative to a model of a stochastic process. That model may be specified directly, often in terms of a small set of parameters, or it may be estimated from data using a variety of learning or optimisation methods. The point here is that contrast information is agnostic to the modelling, estimating, and calculating process. It only measures the information dynamics of a provided model of the process.

Contrast information is formulated as a dynamic information measure since we are concerned with the information dynamics of specific sequences at a specific points in time. This is in contrast to the entropy rate or KS entropy [44], which are global measures. Estimates of KS entropy [49] (approximate entropy, sample entropy, and multiscale entropy) are not only global in nature but are model-free estimators of an information measure. Contrast information is neither an estimator of some other quantity nor model-free; it is a measure that is always relative to a provided model of the sequence. For instance, if two different models are used when calculating contrast information, the same sequence of data will exhibit different information dynamics. Dynamic entropy and adaptive entropy are similar to contrast information in that the temporal aspect takes centre stage in the analysis, but since they are based on Shannon entropy, their analysis must be confined to processes of discrete state spaces, whereas contrast information can work with both discrete and continuous state spaces without the need for quantisation.

In fact, predictive contrast information was already proposed by Abdallah and Plumbley [5]. Their formulation provided the inspiration to develop connective and reflective contrast information, as well as the backward variants in our work. Much of their previous work focused on discrete state spaces [4,56], with a short foray into continuous state spaces [55]. Our work extends theirs, due to not only the other temporal variants of contrast information but also the focus on a continuous time and continuous state spaces.

Closely related to Abdallah and Plumbley’s work is information rate [53], defined as the mutual information between the past and present. Since the information rate is a process measure, it falls into a different category from contrast information. The information rate has multiple equivalent definitions due to identities relating multi-information, mutual information, and entropy. If we convert the information rate from a process measure to a dynamic measure, by moving from a formulation based on mutual information to one based on a specific information, we must remember that those identities no longer hold, so those multiple definitions would no longer be equivalent. But as it stands, the information rate shares many of the same beneficial properties as contrast information, since it works with both discrete and continuous state spaces, and it is coordinate invariant. The core difference is that the information rate is a process measure, not a dynamic measure. A further comparison between predictive contrast information and the information rate is given by Abdallah and Plumbley [5].

Finally, the approach of predictive information of Bialek and Tishby [54] could be incorporated into contrast information, since both measures use the temporal regimes of the past and future. When dealing with stationary processes, varying the position or duration of the past and future regimes within the different temporal variants of contrast information can provide a similar type of analysis to that provided by their predictive information when the context length is varied. Analysing how predictive contrast information of stationary processes behaves as the position or duration of the past or future tend to infinity will also provide a similar analysis as the work of Bialek and Tishby [54], allowing contrast information to also behave as a global measure.

### 8.2. Contributions

We presented a set of novel information measures that are well suited for use in a cognitive model based on the information dynamics of continuous signals, such as our developing IDyOT model. We presented some useful properties of the new measures. We showed that there exists a good correlation, particularly in rank order, between the measures used in the IDyOM system and the discrete version of our measures. Since IDyOM is already empirically demonstrated to be a strong cognitive model, it follows that at least the discrete version of our new measures will serve effectively as a cognitive model also; therefore, it is reasonable to expect that the continuous versions will do the same.

### 8.3. Future Work

The current paper presents contrast information as a measure for the information dynamics of a single stochastic process assuming a single underlying model. Future work will extend contrast information to work with multiple processes and investigate how it can interact with learning or adapting an underlying model. To address how contrast information works with multiple interacting stochastic processes, future work will incorporate the approach of transfer entropy [69] and the related directed information [70], which measure the influence of one stochastic process on another using a particular form of mutual information. In the case of contrast information, we will be interested in how a specific sequences or specific moments in a sequence interact with the information dynamics of another process. To address how contrast information can interact directly with the underlying model, we will follow the approach of the information bottleneck [24], a method for balancing prediction and compression when learning representations of data. Methods such as the deep variational information bottleneck [71] and the conditional entropy bottleneck [72] take a similar approach. The information bottleneck is concerned with finding a quality representation that results in a compressed, but meaningful model of the data. These methods evaluate the quality of compression or prediction using the mutual information between data, output, and embeddings, so we expect that contrast information can be used to measure the information bottleneck of a specific sequence or a specific point in time of a process, thereby connecting it to the representation learning process.

Immediate opportunities for further work in contrast information also include the following:**Boundary Entropy** The first application of the new measures will be in replicating prior segmentation work in music. This will require the adaptation of the information profile peak-picking algorithms for use with contrast information. Subsequently, we will test the continuous measures on speech using the TIMIT dataset (https://paperswithcode.com/dataset/timit; accessed 23 July 2024).**Continuous-state IDyOMS** We will extend our IDyOMS software to allow for continuous states, thus extending its representational reach in music and other domains. This will require substituting the PPM algorithm with models of continuous feature dimensions, as well as methods for combining viewpoint models based on contrast information rather than entropy.**Neural correlates** We aim to collaborate with colleagues in neuroscience to investigate whether and how the neural correlates of perceived sound correspond with IDyOT representations, with the aim to make the system more human-like.**Spectral Knowledge Representation** We will further develop the idea of Spectral Knowledge Representation [40] to allow our system to reason using the symbols identified by segmentation.

In the longer term, we will continue to develop the components required for the IDyOT cognitive architecture, working toward a full implementation.

## Figures and Tables

**Figure 1 entropy-26-00638-f001:**
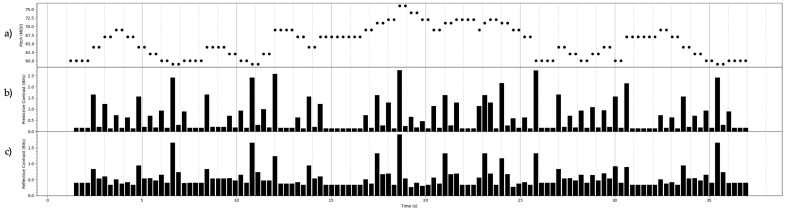
An example of the contrast information associated with a discrete-time, discrete-state stochastic process. In (**a**), a MIDI representation of a folk melody is represented as a discrete-time process through a uniform sampling in time of the data. The discrete state space of the process is inhabited by the MIDI pitch numbers. Note that, though the visualisation represents the pitches as an ordered set, we treat the pitches as an unordered set of discrete states. Through treating all melodies in the corpus as instances of the same stationary first-order DTMC, the maximum likelihood estimate of the transition matrix of the DTMC was found. In (**b,c**), the forward predictive contrast information profile and forward reflective contrast information profile are shown for the melody in (**a**), calculated using the estimated DTMC. Each bar is centred on the present *Y*, where the past *X* is the immediately previous pitch, and the future *Z* is the immediately next pitch. The connective contrast information is omitted here since it is always zero for Markov processes.

**Figure 2 entropy-26-00638-f002:**
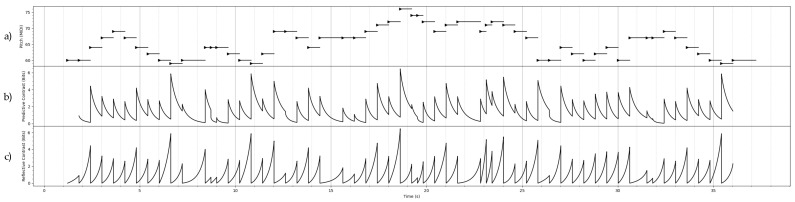
An example of the contrast information associated with a continuous-time, discrete-state stochastic process. In (**a**), a MIDI representation of a folk melody is represented as a continuous-time process, where the onset of a state is indicated by a triangle, and the duration of a state is indicated by the horizontal line. The discrete state space of the process is inhabited by the MIDI pitch numbers. Note that, though the visualization represents the pitches as an ordered set, we treat the pitches as an unordered set of discrete states. Through treating all melodies in the corpus as instances of the same stationary CTMC, the maximum likelihood estimate of the rate matrix of the CTMC was found. In (**b**), the forward predictive contrast information profile is shown. The present *Y* is located at the most recent pitch onset, and the past *X* is located at the pitch onset immediately before *Y*. Each point on the profile curve is centred on the future *Z*, which varies in time from the most recent pitch onset (*Y*) to the next pitch onset, at which point the past and present shift forward to the next pitch onsets. In (**c**), the forward reflective contrast information profile is shown. The past *X* is located at the most recent pitch onset. The future *Z* is located at the next pitch onset. Each point in the profile curve is centred on the present *Y*, which varies in time from the most recent pitch *X* to next pitch *Z*, at which point, the past and future shift forward to the next pitch onsets. The connective contrast information is omitted here since it is always zero for Markov processes.

**Figure 3 entropy-26-00638-f003:**
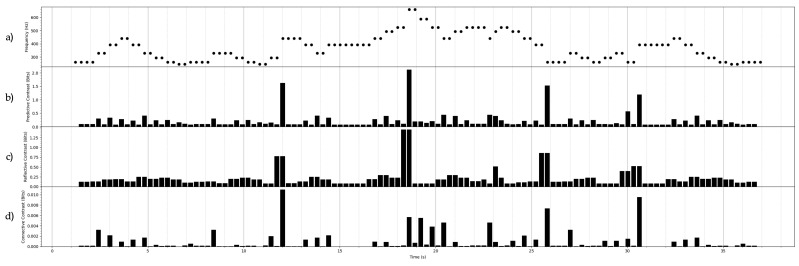
An example of the contrast information associated with a discrete-time, continuous-state stochastic process. In (**a**), a MIDI representation of a folk melody is represented as a discrete-time process through a uniform sampling in time of the data. The continuous state space of the process is inhabited by the frequencies associated with each MIDI pitch. Through treating all melodies in the corpus as instances of the same stationary Gaussian process, the maximum likelihood estimates for the mean and autocovariance of the Gaussian process were found. In (**b**–**d**), the contrast information profiles are calculated using the estimated DTGP. Each bar is centred on the present *Y*, where the past *X* is the immediately previous pitch, and the future *Z* is the immediately next pitch. In this case, using longer duration regimes for the past and future resulted in very similar profiles, so the profiles shown here are generally representative. Note that the scale of the connective contrast is much lower than the other predictive and reflective contrast.

**Figure 4 entropy-26-00638-f004:**
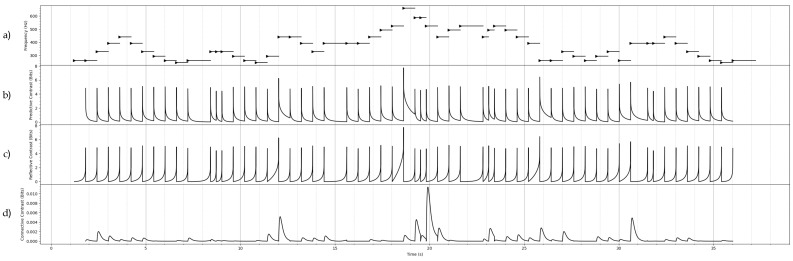
An example of the contrast information associated with a continuous-time, continuous-state stochastic process. In (**a**), a MIDI representation of a folk melody is represented as a continuous-time process, where the onset of a state is indicated by a triangle, and the duration of a state is indicated by the horizontal line. The continuous state space of the process is inhabited by the frequencies associated with each MIDI pitch. Through treating all melodies in the corpus as instances of the same stationary Gaussian process, the maximum likelihood estimates for the mean and autocovariance of the discrete-time Gaussian process were found. The discrete-time autocovariance function was then fit with a high degree polynomial to obtain a continuous-time autocovariance function. In (**b**), the forward predictive contrast information profile is shown. The present *Y* is located at the most recent pitch onset, and the past *X* is located at the pitch onset immediate before *Y*. Each point on the profile curve is centred on the future *Z*, which varies in time from the most recent pitch onset (*Y*) to the next pitch onset, at which point, the past and present shift forward to the next pitch onsets. In (**c**), the forward reflective contrast information profile is shown. The past *X* is located at the most recent pitch onset. The future *Z* is located at the next pitch onset. Each point in the profile curve is centred on the present *Y*, which varies in time from the most recent pitch *X* to the next pitch *Z*, at which point, the past and future shift forward to the next pitch onsets. In (**d**), the forward connective contrast information profile is shown. The past, present, and future follow the same scheme as described for forward predictive contrast information in (**b**). Note that the scale of the connective contrast is much lower than the other the predictive and reflective contrast.

**Table 1 entropy-26-00638-t001:** Sequence notation schemes for sequences of random variables.

Regime	Continuous	Notation	Discrete	Notation
Past	$\left\{S(\tau ):\tau <t\right\}$	*X*	$\left\{\ldots ,S_{n-2},S_{n-1}\right\}$	*X*
Near Past	$\left\{S(\tau ):t-u\leq \tau <t\right\}$	$X^{u}$	$\left\{S_{n-j},\ldots ,S_{n-1}\right\}$	$X^{j}$
Point Past	S(t−u)	$X_{u}$	$S_{n-j}$	$X_{j}$
Present	S(t)	*Y*	$S_{n}$	*Y*
Point Future	S(t+v)	$Z_{v}$	$S_{n+k}$	$Z_{k}$
Near Future	$\left\{S(\tau ):t<\tau \leq t+v\right\}$	$Z^{v}$	$\left\{S_{n+1},\ldots ,S_{n+k}\right\}$	$Z^{k}$
Future	$\left\{S(\tau ):\tau >t\right\}$	*Z*	$\left\{S_{n+1},S_{n+2},\ldots \right\}$	*Z*

**Table 2 entropy-26-00638-t002:** Correlation between information content and forward predictive contrast information. Zeroth-order model (i.e., constant distribution) is included to emphasise the difference between this and information dynamic models.

	CPITCH		DUR		CPITCH × DUR	
Order	ρ	$r_{s}$	ρ	$r_{s}$	ρ	$r_{s}$
*0*	*.86*	*.88*	*.89*	*1*	*−.83*	*−.82*
1	.85	.86	.72	.51	.34	.38
2	.76	.77	.70	.66	.45	.47
3	.73	.76	.70	.69	.23	.28
4	.75	.81	.71	.72	.19	.33
5	.77	.86	.73	.78	.23	.39
6	.78	.88	.74	.80	.32	.45
7	.80	.90	.75	.82	.45	.50
8	.82	.91	.75	.84	.58	.54
9	.84	.92	.76	.86	.66	.57
10	.85	.92	.77	.87	.69	.60

**Table 3 entropy-26-00638-t003:** Correlation between entropy and expected forward predictive contrast information. Zeroth-order model (i.e., constant distribution) is included to emphasise the difference between this and information dynamic models.

	CPITCH		DUR		CPITCH × DUR	
Order	ρ	$r_{s}$	ρ	$r_{s}$	ρ	$r_{s}$
*0*	*1*	–	*1*	*–*	*1*	*–*
1	.20	−.06	.36	−.48	−.28	−.23
2	.24	.28	.31	−.14	.06	.33
3	.51	.50	.40	.29	.27	.36
4	.67	.68	.44	.29	.34	.46
5	.73	.81	.50	.42	.37	.53
6	.75	.89	.55	.49	.42	.59
7	.77	.93	.60	.60	.54	.64
8	.80	.95	.64	.70	.68	.68
9	.83	.96	.66	.76	.77	.71
10	.85	.97	.67	.80	.81	.73

## Data Availability

The data used for the experiment reported in Section 6 may be found in KERN** format in the HumDrum resource at https://kern.humdrum.org/cgi-bin/ksbrowse?type=collection&l=/users/craig/songs/creighton/nova, accessed on 23 July 2024. The data was converted to MIDI format, as part of earlier work [2], using Marcus Pearce’s IDyOM system, available at https://github.com/mtpearce/idyom/, accessed on 23 July 2024. The IDyOMS system used in the experiment may be found at https://github.com/nick-harley/Idyoms, accessed on 23 July 2024.

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
