# Peer review of "Contrast Information Dynamics: A Novel Information Measure for Cognitive Modelling"

_entropy, 2024, doi:10.3390/e26080638_

Round 1

Reviewer 1 Report

Comments and Suggestions for Authors

The paper introduces a set of information theoretic measures applied to discrete and continuous musical signals. The results and conclusions are based on analysis of few musical examples, showing different correlations between the proposed information measures for three types of musical features or viewpoints.

The overall conclusion is that the proposed contrast information does not significantly differ from previous measures by the same authors, so the utility of the work remains questionable. To be significant, the experiments should be extended to larger set of examples or provide a human expert analysis.

The majority of the paper deals with formal definition of the contrastive information measures, which seems to be a variant of previous measures of predictive information referencing back to the authors' earlier IDyoM work. The paper is largely omitting or superficially mentioning other work in the field like Information Rate by Dubnov, Predictive Information by Bialek, Transfer Entropy by Schreiber, Information Bottleneck by Tishby and its variants (Alemi and Fischer's Conditional Entropy Bottleneck), Directed Information by Massey. Such works seem to be conceptually related to this research, also from the broader cognitive perspectice, and should be considered in comparison to the proposed measure.

In terms of viewpoints and musical dynamics models and estimators, it would be good to consider recent related works using deep neural network for learning music representation and generative AI models, such as Deep Music Information Dynamics that seems to be directly relevant to the paper.

Author Response

Please see PDF file.

Reviewer 2 Report

Comments and Suggestions for Authors

I consider that the submission is interesting. However, I'd like to comment

the following: 

1. The idea the authors start with is Shannon entropy. There are many publications where this concept evolved (Aprox. Entropy, Sample Entropy, 

Mutiscale Entropy, Fuzzy Entropy...).  Particularly, ApEn  was improved by SaEn as a result of being a biased estimator. What can you say abt this in your case?

2. Besides (1), those entropies have had (time series) length-based problems. In your corollary 1, you show that as the mean absolute error (MAE) between information content â„“(b|c) and contrast information I [A : b | c] is a monotonically decreasing function of the length of A and the length of C with a maximum possible value of H(B).  What happens for very long time series? (in order of millions samples) but also in contrast, what happens when a time series is very short? (less than 2000 or 1000 samples). 

3. What is the computation time for a "medium" long series, a very short one and a very long TS? See (2). 

4. Which type of computational errors exist for very short and very long TS?

5. What is the algorithmic complexity of the whole algorithm? Compute it. 

6. What happens if the stochastic processes you consider are not necessarily Gaussian nor stationary?

7. Moreover, what happens if they are heteroskedastic? 

8. What is the time/algorithmic complexity of your algorithm in case of dealing with multivariable TS? Is it reliable?

9. Compare your algorithm with some other known (and non problematic) entropy. For instance SaEn, MSE, Composite MSE ... And some other new strategies I found out in Internet: Fuzzy Entropy (several types), Dynamic Bivariate Entropy, Adaptive Entropy...). Please cite those works and provide advantages/disadvantages of them versus yours. 

10 . Does the variance of the time series vary? This is an issue in ApSa, SaEn...

Please include all these issues in your report.

Author Response

Please see PDF document.

Reviewer 3 Report

Comments and Suggestions for Authors

This is an interesting and very promising paper with high academic standards. The technicality of the description, however, is extremely challenging. This means that the major strength of this paper is also its major weakness: readability and understandability. If the paper aims at a target readership with sufficient technical background knowledge of signal analysis, the paper could be welcomed rather easily. The paper, however, is also about music, which means that music scientists may also be interested in reading and understanding the paper. In its current form, the paper is too technical, and is not understandable, even for readers with a lot of mathematical and statistical training. Some major modifications must be done, therefore to make the contents more understandable. I list below some possible hints to improve the readability. Besides this problem of understandability, the paper deals with very important and challenging topics. The major emphasis on information dynamics and the continuous description of the signal is a very important approach that deserves wider distribution in music studies. In order to be as constructive as possible, I list below some general remarks and detailed comments. 

General remarks 

·      The English language use is OK and is quite idiomatic. 

·      The style of writing is quite mature but is too cryptic. Besides the elaborated technical descriptions, there is need of more intuitive description of some focal concepts and analytical approaches, so that common readers may understand at least the major insights behind the technical stuff. This is actually rather problematic, and much rewording is needed to make the paper more understandable. 

·      There is a possible suggestion (not mandatory) to split the paper in a rather readable part and a more technical part (appendix) that elaborates in more technical terms. There is also the possibility of inserting a box with an overview (definition and short intuitive description) of the newly introduced conceptual terms. This could help a lot to improve the readability. 

·      The mathematical formulas and derivations are very difficult to understand for common readers. There are multiple abbreviations which are not explained and which leave the reader with a feeling of helplessness. Some more explicit explications of the distinct steps is needed. 

·      Some technical terms are introduced, but are not explained at “first” appearance. In the case that they are explained later in the text, there should be at least be some reference to this explanation (e.g. “see below”), but it is recommended to provide a “short” and “intuitive” description at first appearance. This holds, e.g., for coordinate invariance, parameterization invariance, KL divergence, Markov and Gaussian processes, determinant of the Jacobian. This is all very technical stuff for common readers, even if some of it may seem to be common knowledge for the trained scholars and of course for the authors of this paper. 

·      After reading the whole paper, the idea of contrast information is not yet sufficiently clear to have it as a take home message. A more intuitive description is needed at first introduction of the term. The concept, however, is very interesting and promising.

·      The contents as a whole, are very interesting. This hold especially for the cognitive modelling of sequential perception of music, which is very needed. Also, the concept of information dynamics is an important new introduction to the field. The change of information from one moment to the next and the focus on specific instantiations of stochastic processes seem to be a very welcomed scientific addition to the philosophical theories of the constitution of time (e.g., the distinction between retention, now moment, and protention in the phenomenological philosophy by Husserl). Some positioning of the newly introduced applications in this paper against older theories of time should improve the impact of the paper considerably. 

·      There are lot of claims in this paper, which are wrapped up in mathematical descriptions, but even if the formalism of the definitions seems to be OK, questions can be raised as to the empirical verification of these claims. As such, they can seem to be some arbitrary or unmotivated, as they lack connection with empirical studies about time perception and cognition. 

Detailed comments

·      Title: Perhaps the title should be extended by adding “in language and music”

·      Lines 9 ff: very strong introduction. Good statement of the major claims of the paper.

·      Line 13: to make the paper more readable and understandable, do not be afraid to add a short intuitive description of the term stochastic, even if this is common knowledge for many of us. Proposal:  … of stochastic processes, which are characterized by a sequence of random variables indexed by time.

·      Line 34: provide a short intuitive description of the concepts of coordinate invariance and parameterization invariance or add “see below for explanation of the terms” between brackets. Bot terms are explained later in the paper, but the reader should be informed already at first appearance of the term. 

·      Line 36: same remark for the term “relative entropy”, which is not common knowledge for the common readership of the journal. This holds even more for the term KL divergence, which is very technical. A possibility could be: Kullback-Leibner (Kl) divergence (a type of statistical distance to measure howe one probability distribution is different from a reference probability distribution). Such short rather intuitive description can be very helpful for the non-trained reader to grasp the meaning of the text. 

·      Lines 52 ff: the distinction between predictive, connective, and reflective contrast is very important for the paper. The concepts, however, are quite abstract, and are not sufficiently explained here to grasp their meaning. Please provide a more intuitive description to help the reader. 

·      Line 95: the principle of free energy should be explained also in more intuitive terms: what does the word “free” stand for?  Something like: A measure of surprise or uncertainty that describes the objective of an agent to maintain a stable preferred state and minimize the uncertainty about the environment??

·      Line 120: the IDyOM model seems to be reductionistic by reducing the measurements to “pitch”. This should be approached in more critical way. Music is more than pitches, especially in case of electronic music, musique concrete, noise music, etc. This reduction should also be discussed in the conclusions section. An alternative for pitch could be the ecological concept of “event” is a more generic concept.

·      Line 142: explain more intuitively and shortly the meaning of boundary entropy. All these technical terms are quite informative and very handy to provide operational definitions, but they should not be taken for granted as common knowledge by the readers.

·      Lie 174: I wonder whether some digression about the concept of musical entrainment could have a place here?

·      Line 192: opening up here the reference to Hilbert space of resonances seems to be a very promising avenue for future research. The concept, however, is extremely technical and should be explained a little in intuitive terms, even if this is a quite impossible task. The aim must be to give the reader an idea of what this paragraph is about and also to avoid the danger of pedantry. 

·      Line 219: here the concepts of coordinate invariance and parameterization are explained in quite understandable terms. This should be done, however, at first appearance of the terms. 

·      Line 249: the mathematical formulas and derivations are well-constructed but are very technical. Most readers will not be able to follow. It is very important to clearly explain all abbreviations at first appearance. What does Cstand for? Something about context?

·      Line 263: the concept of determinant of the Jacobian may be something totally unknown for most readers. Explain somewhat more in intuitive term, or explain at least why this technique is used. 

·      Line 268:  probability is always zero: why? All these quasi-trivial evidences (for the authors) are quite entropic for common readers and they leave them with a feeling of lack of mastery and not-understanding. There is a danger that they will stop reading here, especially is they are not trained statisticians of mathematicians. 

·      Lie 278: not coordinate invariant. Why? All these statements are stated as being evident, but again, this is not obvious for common readers. Much more help is needed to make things understandable. 

·      Lie 279 ff: from here, most readers will no longer be able to follow. Much too technical for common readers. Is there a possibility to make a distinction here between the major steps of reasoning, the logic behind the development of ideas and the mathematical elaboration, and to provide the more technical part at the end of the aper in a technical appendix? It is clear that the high-level of mathematical competence of the authors obscures their valuation of what common readers can understand. Much depends also on the kind of journal and the standards of the journal, but given that this paper is also about music, the contents must be presented in a more understandable way. 

·      Line 279: why introducing the logarithm here? Shannon entropy? Explain a little. 

·      Line 287: differential entropy is not coordinate invariant. Why? There is always a lack of motivation in these statements.  Readers with experience in empirical research may find this a bit arbitrary and unmotivated (though it surely is not).

·      Line 290: wat does E stand for? (expectation?) Please help the reader at least a little. 

·      Lie 316: the concept of index set seems to have a lot operational power. Please a little more in detail the relevance of this concept. 

·      Line 419: all these proofs are extremely technical and will be understood only by a very selected sample of the readership. They make the text very hard to read. Is it necessary to insert them in the main text? Can they be moved to the end of the paper in a kind of technical appendix? This is just a selection. 

·      Lines 457 ff: this is very interesting stuff. It seems to provide to some extent an operational description of the phenomenological description of time constitution of time, with major concepts as retention, now moment, and protention. Perhaps a reference to this philosophical background should be made here, if possible. 

·      Line 509: here also an example of a very successful intuitive description of reflective contrast information. This should have been done earlier in the paper to help the reader to understand the development of thoughts. 

·      The following pages and figures are extremely difficult to understand. Most readers will not continue reading and will miss the major take home of the paper. 

Author Response

Please see PDF document.

Round 2

Reviewer 1 Report

Comments and Suggestions for Authors

Thank you to the authors for addressing the previous comments. It significantly improved the readability and completeness of the paper. There are still some significant issues that need to be addressed before publishing. The emphasis on contrast information being coordinate invariant does not pertain to the novelty of the proposed measure, as mutual information itself is coordinate invariant. Conditioning of mutual information on context does not establish the coordinate invariance. Please clarify. 

The use of specific and contrastive information still seems to be closely a derivative of Plumbley's PID. To make the definitions and novelty clear, I propose to provide the mathematical expressions for definitions of other existing measures such as information rate, predictive information and predictive information rate to compare and explain how the contrast and specific information differ or supplement them. 

In 4.2, the dual notation of mutual information with square brackets versus round parentheses is unclear. Please explain or omit this difference, since the change in types of parentheses is not very noticeable and needs to be pointed out, if this is needed.  

In the appendix, change "Theorem" to Corollary, since these are not theorems of proofs, but statement of resulting properties.  

About the results, my concern is that the melodic analysis and statistics are still using a discrete data, which does not incorporate or justify the novelty of contrast information for continuous variables. Can you actually run experiments on continuous variables such as audio feature vectors to include aspects of timbre or other musically significant parameters? There seems to be a disconnect between the stated novelty of the proposed mathematical measures and the type of data used in the analysis.

The example in Figure 3 is using a continuous representation of frequencies, but the actual data are discrete. If this was a continuous pitch profile, that would be more appropriate examples of a continuous data. As mentioned above, other audio features would be more natural candidates for such demonstration.

Please provide the reference to the corpus of 152 folk melodies you used. It would also be helpful to make the results reproducible by providing the implementation of the estimators, including the PPM used for the sequence modeling of different order.

Since cognitive experiments are not part of the paper, I would suggest omitting the mention of cognitive modeling from the title and presenting this as an extension of the field of information dynamics. 

Author Response

Please see PDF file.

Reviewer 2 Report

Comments and Suggestions for Authors
  1.  Please add the references I requested when I asked to compare your entropy versus: Adaptive Entropy, Bivariate Entropy, Composite Entropy, Dynamic Bivariate Entropy (DREES).  
  2. Besides the examples you provide in the manuscript, in which other real life situations could your algorithm be applied?

Author Response

Please see PDF file

Reviewer 3 Report

Comments and Suggestions for Authors

Dear authors,

The resubmitted version of your paper is much improved compared to the first version. Thanks for this. The readability and understandability of the paper is much better now, though the technicality of the contents is still very high. But perhaps the readership of the journal is sufficiently trained to cope with this style of reporting. I have the impression that most of my comments and remarks have been addressed to some extent, which means that I am wiling to accept the paper for publication. Careful rereading is still needed, however, as there are some minor typos, but there is also still a lot of redundancy in the newly added text. Making some of these additional explanations more concise could still improve the style of writing.

Author Response

Please see PDF file.

Round 3

Reviewer 2 Report

Comments and Suggestions for Authors

There is one comment that has not been addressed.

" Besides the examples you provide in the manuscript, in which other real life situations could your algorithm be applied?

Any phenomenon that can be modeled as a stochastic process or dynamical system can use contrast information to measure information dynamics. Some obvious areas include signal processing, machine learning, finance, logistics, healthcare telemetry, and so much more!"

Please ask them to do it.